# Physiological potential and evolutionary trajectories of syntrophic sulfate-reducing bacterial partners of anaerobic methanotrophic archaea

Ranjani Murali[1]*, Hang Yu[2,3], Daan R. Speth[1,4], Fabai Wu[5], Kyle S. Metcalfe[6], Antoine Crémière[2], Rafael Laso-Pèrez[7¤], Rex R. Malmstrom[8], Danielle Goudeau[8], Tanja Woyke[8], Roland Hatzenpichler[9], Grayson L. Chadwick[2,6], Stephanie A. Connon[2], Victoria J. Orphan[1,2]*

1 Division of Biology and Biological Engineering, California Institute of Technology, Pasadena, California, United States of America, 2 Division of Geological and Planetary Sciences, California Institute of Technology, Pasadena, California, Unites Stated of America, 3 Department of Physics and Astronomy, University of Southern California, Los Angeles, California, United States of America, 4 Division of Microbial Ecology, Centre for Microbiology and Environmental Systems Science, University of Vienna, Vienna, Austria, 5 ZJU-Hangzhou Global Scientific and Technological Innovation Center, Zhejiang University, Hangzhou, China, 6 Department of Plant and Molecular Biology, University of California, Berkeley. Berkeley, California, United States of America, 7 Systems Biology Department, Centro Nacional de Biotecnología (CNB-CSIC), Madrid, Spain, 8 DOE Joint Genome Institute, Department of Energy, Berkeley, California, United States of America, 9 Department of Chemistry and Biochemistry, Montana State University, Bozeman, Montana, United States of America

¤ Current address: Biogeochemistry and Microbial Ecology Department, Museo Nacional de Ciencias Naturales (MNCN-CSIC), Madrid, Spain
* m.ranjani@gmail.com (RM); vorphan@caltech.edu (VJO)

## Abstract

Sulfate-coupled anaerobic oxidation of methane (AOM) is performed by multicellular consortia of anaerobic methanotrophic archaea (ANME) in obligate syntrophic partnership with sulfate-reducing bacteria (SRB). Diverse ANME and SRB clades co-associate but the physiological basis for their adaptation and diversification is not well understood. In this work, we used comparative metagenomics and phylogenetics to investigate the metabolic adaptation among the 4 main syntrophic SRB clades (HotSeep-1, Seep-SRB2, Seep-SRB1a, and Seep-SRB1g) and identified features associated with their syntrophic lifestyle that distinguish them from their non-syntrophic evolutionary neighbors in the phylum Desulfobacterota. We show that the protein complexes involved in direct interspecies electron transfer (DIET) from ANME to the SRB outer membrane are conserved between the syntrophic lineages. In contrast, the proteins involved in electron transfer within the SRB inner membrane differ between clades, indicative of convergent evolution in the adaptation to a syntrophic lifestyle. Our analysis suggests that in most cases, this adaptation likely occurred after the acquisition of the DIET complexes in an ancestral clade and involve horizontal gene transfers within pathways for electron transfer (CbcBA) and biofilm formation (Pel). We also provide evidence for unique adaptations within syntrophic SRB clades, which vary depending on the archaeal partner. Among the most widespread syntrophic SRB, Seep-SRB1a, subclades that specifically partner ANME-2a are missing the cobalamin synthesis pathway,

**Data Availability Statement:** All the metagenome assembled genomes (MAGs) created for this study, along with associated metadata and biosample data are available from the NCBI database (BioProject: PRJNA762493). Metagenome assemblies from sorted aggregates that we used here are made available on the IMG database under the following IDs – 3300036218, 3300036221, 3300036226, 3300036304, 3300036329, 3300036259. The original tree files have been made available on the first author's Github page (https://github.com/ranjani-m/syntrophic-SRB) and they have also been uploaded as part of online supplementary data. All other relevant data is provided as supplementary material in this manuscript.

**Funding:** This research was supported by the Gordon and Betty Moore Foundation Models in Marine Symbiosis (GBMF Grant #9324, to V.J.O), the U.S. Department of Energy BER(Award Number: DE-SC0022991 to V.J.O); and a FICUS grant (Award doi: 10.46936/fics.proj.2017.49956/60006219 to V.J.O.) through the U.S. Department of Energy Joint Genome Institute (https://ror.org/04xm1d337), a DOE Office of Science User Facility supported by the Office of Science of the U.S. Department of Energy operated under Contract No. DE-AC02-05CH11231). Deep sea samples for this project were collected in part through support from the National Science Foundation (OCE-1634002, to V.J.O) and the Science and Technology Center for Dark Energy Biosphere Investigations (C-DEBI, OCE-0939564, to V.J.O). R.L.-P. was funded by a Juan de la Cierva grant (FJC2019-041362-I) from the Spanish Ministerio de Ciencia e Innovación and by a Ramón y Cajal grant (RyC2021-031775-I) from the Spanish Ministerio de Ciencia e Innovación (MCIN/AEI/10.13039/501100011033) and the European Union («NextGenerationEU»/PRTR). The funders had no role in study design, data collection and analysis, decision to publish, or preparation of the manuscript

**Competing interests:** The authors have declared that no competing interests exist.

**Abbreviations:** ANI, average nucleotide identity; ANME, anaerobic methanotrophic archaea; AOM, anaerobic oxidation of methane; BIC, Bayesian information criterion; BONCAT, BioOrthogonal Non-Canonical Amino Acid Tagging; DIET, direct interspecies electron transfer; EET, extracellular electron transfer; eCIS, extracellular contractile injection system; FACS, fluorescence-activated cell sorting; FBEC, flavin-based electron confurcation; FISH, fluorescence in situ hybridization; IMG, integrated microbial genomes; MAG, metagenome-assembled genome; MDA, multiple displacement

suggestive of nutritional dependency on its partner, while closely related Seep-SRB1a partners of ANME-2c lack nutritional auxotrophies. Our work provides insight into the features associated with DIET-based syntrophy and the adaptation of SRB towards it.

## Introduction

Syntrophy is metabolic cooperation between microorganisms for mutual benefit. It is a common adaptation in low-energy environments and enables the utilization of substrates which neither organism could metabolize on its own [1,2]. While the driving force for different microbial syntrophic interactions may vary, both partners benefit from sharing nutrients and electrons in this way, combining their resources and avoiding the need for both partners to expend energy for the synthesis of common nutrients [1,3]. Syntrophic interactions appear to be specific in at least some cases, with the same organisms co-associating across different ecosystems and environments [4]. However, we do not yet understand the physiological basis driving the specificity of interactions, often because syntrophic associations are difficult to grow in culture. Characterizing the specificity of these interactions is challenging with uncultured syntrophic consortia in the environment [2]. A classic syntrophic partnership is at the heart of the important biogeochemical process, sulfate-coupled anaerobic oxidation of methane (AOM) [5]. Anaerobic methanotrophic archaea (ANME) and sulfate-reducing bacteria (SRB) coexist in multicellular consortia, with ANME performing methane oxidation coupled to sulfate reduction by the SRB [6–8]. Direct interspecies electron transfer (DIET) from ANME to SRB is predicted to be the dominant mechanism of syntrophic coupling in many observed cases of sulfate-coupled AOM [9,10] though diazotrophic nitrogen is also shared between these partners [11–13]. There is rich ecological diversity in the observed examples of AOM with taxonomically divergent groups of ANME coexisting with an equally diverse group of SRB in consortia that appear morphologically different [13,14]. ANME-SRB consortia exist in hydrothermal vents [15], in cold-seeps [6,7,16], in mud volcanoes [17], and a euxinic basin [18], and can form tight spherical aggregates [16,19], dense microbial mats [20], or loose associations [19,21]. Past work has suggested that some ANME-SRB associations are more specific than others [13,22,23], and ecophysiological studies have demonstrated differences in genes expressed, even pertaining to DIET [24,25]. To investigate whether there is underlying structure to this variety of ANME-SRB interactions in different ecosystems, and to infer the evolutionary trajectories that led to these extant phenomena, it is important to establish a taxonomic, ecological, and physiological framework within which to organize our observations.

Investigation of the archaeal and bacterial lineages involved in AOM identified at least 3 divergent taxonomic groups of archaea, by analysis of 16S rRNA gene sequences and fluorescence in situ hybridization (FISH)–ANME-1 (Methanophagales), ANME-2, and ANME-3 (Methanovorans) [6,17,26]. All 3 of these groups are clades within the phylum Halobacterota, and ANME-2 is subdivided into the clades ANME-2a (*Methanocomedenaceae*), ANME-2b (*Methanomarinus*), ANME-2c (*Methanogasteraceae*), and ANME-2d (*Methanoperedenaceae*) [14]. In a recent paper [14], Chadwick and colleagues established a robust taxonomic framework for ANME, identified key biochemical pathways in the archaea that are important for AOM and demonstrated that the capability for extracellular electron transfer (EET) is a significant metabolic trait that differentiates ANME from its nearest evolutionary neighbors that are typically methanogens or alkane oxidizing microorganisms [27–29]. Within the ANME

amplification; MIET, mediated interspecies electron transfer; PCA, principal component analysis; PGAP, Prokaryotic Genome Annotation Pipeline; PLTS, phage-like translocation system; pmf, proton motive force; rTCA, reductive tricarboxylic acid cycle; SRB, sulfate-reducing bacteria.

(ANME-1, ANME-2a, ANME-2b, ANME-2c, and ANME-3) that are known to partner with SRB [6,16,19,26,30], they were also able to show that there were differences between clades with respect to putative EET pathways, electron transport chains, and biosynthetic pathways [14]. Analysis of ANME genomes and inferences from previous physiological data [24,25] also suggested differences in secretion machinery and cellular adhesion proteins in the archaeal partner that might affect syntrophic interactions with the partner bacteria [14]. The goal of this work is to establish a similar framework for the well-established clades of sulfate-reducing partner bacteria, to identify important differences in biochemical pathways, and to identify traits that might correlate with differences in ANME-SRB partnership pairing.

The key to understanding the evolutionary diversification of SRB is to understand the taxonomic background of each syntrophic SRB involved in this process, since taxonomy is the "expression of evolutionary arrangement" [31] and the phenotypic traits that differentiate the syntrophs from their non-syntrophic evolutionary neighbors. Previous work demonstrated that there were 4 sulfate-reducing bacterial clades within the Desulfobacterota that partner ANME–HotSeep-1 [24,32], Seep-SRB2 [24], Seep-SRB1a [25,33], and Seep-SRB1g [13,33]. Other bacteria and archaea (seepDBB within the *Desulfobulbaceae* [34], alpha- and beta-proteobacteria [35] and verrucomicrobia [36], Anaerolineales and Methanococcoides [37]) have also been observed to associate with ANME. However, in this work, we investigated only these 4 clades (HotSeep-1, Seep-SRB2, Seep-SRB1a, and Seep-SRB1g) that are consistently and most often found in association with ANME across different ecosystems [23]. With the use of publicly available datasets and 15 metagenomes, we generated in this study from different marine ecosystems (seeps located off the coast of Costa Rica and off the coast of S. California, as well as hydrothermal vents in the Gulf of California), we curated a database of 46 syntrophic SRB metagenomes with multiple representatives from each clade. With the use of the Genome Taxonomy Database [38], we created a taxonomic framework to reproducibly classify these syntrophic SRB and proposed scientific names according to the latest guidelines. Significantly, our curated database of representative genomes and 16S rRNA sequences would allow future studies to differentiate the known syntrophic Seep-SRB1a and Seep-SRB1g clades from the non-syntrophic members (Seep-SRB1b, Seep-SRB1c, Seep-SRB1d, Seep-SRB1e, and Seep-SRB1f) of the polyphyletic clade Seep-SRB1 [23,39].

To differentiate the phenotypic traits of syntrophic SRB from non-syntrophic SRB, we synthesized information from prior physiological experiments [24,25] and provide a detailed biochemical description of pathways that are necessary for the formation of ANME-SRB consortia. Our analysis demonstrated that the syntrophic SRB contain all the genomic traits consistent with their participation in DIET (including EET pathways), and with the formation of a multispecies conductive biofilm (cellular adhesion pathways, polysaccharide biosynthesis pathways). Comparative genome analysis between syntrophic genomes and over 550 non-syntrophic bacteria within the phylum Desulfobacterota, showed that these traits are rare in non-syntrophic SRB. We also investigated the importance of partner-pairing as a meaningful ecological factor that differentiates species of syntrophic SRB. We tested this by sequencing single ANME-SRB consortia, isolated by fluorescence-activated cell sorting (FACS) [36]. We showed that Seep-SRB1a partners of ANME-2c appear to have cobalamin biosynthesis pathways while Seep-SRB1a partners of ANME-2a do not, indicating the latter species of Seep-SRB1a had developed a nutritional dependence on its partner. These results indicate that there might be characteristics that are unique to different ANME-SRB pairings and lay the groundwork for future studies to use a species-level partnership framework to explore the co-diversification of ANME and SRB. Our study highlights the complex evolutionary trajectory of adaptation of these SRB to syntrophy with ANME and provides insight into the defining features of DIET-based syntrophic interactions.

## Results and discussion

### Taxonomic diversity within syntrophic SRB partners of methanotrophic ANME

To investigate the adaptation of SRB to a partnership with ANME, we first placed them into their taxonomic context and assessed the phylogenetic diversity within the SRB clades (Seep-SRB1a, Seep-SRB1g, Seep-SRB2, and HotSeep-1). For this analysis, we compiled a curated dataset of metagenome-assembled genomes (MAGs) from these SRB clades including 34 previously published genomes [25,32,33,40–44] and 12 MAGs assembled for this study. Five of these genomes were reconstructed from seep samples collected off the coast of California, Costa Rica, and within the Gulf of California. We also sequenced single ANME-SRB consortia that were sorted by FACS after they were SYBR-stained as previously described [37]. With this technique, we could be confident of the assignment of partners that physically co-associate within the sequenced aggregates and begin to identify partnership-specific characteristics. From sequencing of single consortia, we obtained 2 genomes of ANME-2b associated Seep-SRB1g, 1 genome of ANME-2a associated Seep-SRB1a, and 3 genomes of ANME-2c associated Seep-SRB1a (Table 1). We recovered an additional 3 genomes of the nearest evolutionary neighbors of HotSeep-1 within the order Desulfofervidales since this order of bacteria is very poorly represented in public databases. Our dataset for comparative genomics analysis comprised the above mentioned 46 genomes of syntrophic SRB and over 550 other bacteria from Desulfobacterota. Having compiled this dataset of syntrophic SRB, we also designated type material and proposed formal names for 3 of the syntrophic SRB clades, Seep-SRB2 (*Candidatus* Desulfomithrium gen. nov.), Seep-SRB1a (*Candidatus* Syntrophophila gen. nov.), and Seep-SRB1g (*Candidatus* Desulfomellonium gen. nov.). The genomes designated as type material are identified in Figs 1 and S1. Further details are available in the Supporting information as a proposal for formal nomenclature for Seep-SRB1a, Seep-SRB1g, and Seep-SRB2 (S1 Text).

Details for the phylogenetic placement of each of these clades using 16S rRNA phylogeny, concatenated ribosomal protein phylogeny, and the Genome Taxonomy Database are provided in Materials and methods and S1–S3 Figs. HotSeep-1 is a species within the order Desulfofervidales, an order that is largely associated with thermophilic environments (with 1 exception, Desulfofervidales sp. DG-60 was sequenced from the White Oak Estuary [46]). Members of HotSeep-1 are the best characterized members of this order and are known to be syntrophic partners to thermophilic clades of methane-oxidizing ANME-1 [14,24] as well as alkane-oxidizing archaeal relatives "*Candidatus* Syntrophoarchaeum butanivorans," "*Candidatus* Syntrophoarchaeum caldarius" [27], and ethane-oxidizing "*Candidatus* Ethanoperedens thermophilum" [40]. Seep-SRB2 is a genus-level clade within the order Dissulfuribacterales [47–49] and class Dissulfuribacteria. Dissulfuribacterales include the genera Dissulfuribacter and Dissulfurirhabdus [47–49], which are chemolithoautotrophs associated with sulfur disproportionation. Seep-SRB1g is a species level clade which groups within a taxonomic order that also includes Seep-SRB1c (Fig 1 and Table 1). This order falls within the class Desulfobacteria along with the sister order Desulfobacterales. Like the Desulfofervidales, the order with Seep-SRB1g is poorly characterized, yet its most well-described members are the Seep-SRB1g that are obligate syntrophic partners of ANME, accepting electrons from the archaeal partner to reduce sulfate [13,33]. Seep-SRB1a is a genus-level clade that along with the genus Eth-SRB1 forms a distinct family within the order Desulfobacterales (Figs 1 and S1 and S2 Table). Many of the well-characterized members of Desulfobacterales such as *Desulfococcus oleovorans*, *Desulfobacter hydrogenophilus*, *Desulfosarcina* BuS5 are known as hydrogenotrophs and hydrocarbon degraders [50–52]. The nearest evolutionary relative of Seep-SRB1a are the Eth-SRB1 first characterized as a syntrophic partner of ethane-degrading archaea [29]. Seep-SRB1a

**Table 1. List of genomes from syntrophic SRB labeled by clade, generated in this study and compiled from public databases.**

| Assembly | Organism | Proposed formal name | Genome size (bp) | Completeness | Contamination | GTDB_classification | Geographic location | Reference |
|---|---|---|---|---|---|---|---|---|
| JAJSZM000000000 | Desulfofervidales_sp._FWG156* | | 1452671 | 90.31 | 1.42 | d__Bacteria;p__Desulfobacterota;c__Desulfofervidia;o__Desulfofervidales; f__DG-60;g__;s__ | Pescadero Basin, Gulf of California | This study |
| JAJSZL000000000 | Candidatus Desulfofervidaceae sp. 1* | | 2230686 | 97.97 | 3.46 | d__Bacteria;p__Desulfobacterota;c__Desulfofervidia;o__Desulfofervidales; f__Desulfofervidaceae;g__;s__ | Pescadero Basin, Gulf of California | This study |
| JAJSZS000000000 | Candidatus Desulfofervidaceae sp. 2* | | 2268002 | 98.24 | 3.49 | d__Bacteria; p__Desulfobacterota; c__Desulfofervidia; o__Desulfofervidales; f__Desulfofervidaceae; s__ | Hydrothermal vent Pescadero Basin, Gulf of California | This study |
| GCA_001577525.1 | HotSeep1 | | 2540211 | 96.75 | 4.27 | d__Bacteria; p__Desulfobacterota; c__Desulfofervidia; o__Desulfofervidales; f__Desulfofervidaceae; g__Desulfofervidus; s__Desulfofervidus auxilii | Hydrothermal vent Guaymas Basin, Gulf of California | [32] |
| HotSeep1_draft_B50 | HotSeep1 B50 | | 2215853 | 95.53 | 9.35 | d__Bacteria; p__Desulfobacterota; c__Desulfofervidia; o__Desulfofervidales; f__Desulfofervidaceae; g__Desulfofervidus; s__Desulfofervidus auxilii | Hydrothermal vent Pescadero Basin, Gulf of California | This study |
| CAJIMJ000000000 | HotSeep1 E37 | | 1863098 | 87.40 | 3.46 | d__Bacteria; p__Desulfobacterota; c__Desulfofervidia; o__Desulfofervidales; f__Desulfofervidaceae; g__Desulfofervidus; s__Desulfofervidus auxilii | Guaymas Basin, Gulf of California | [40] |
| CAJIMK000000000 | HotSeep1 E50 | | 1842718 | 70.82 | 5.08 | d__Bacteria; p__Desulfobacterota; c__Desulfofervidia; o__Desulfofervidales; f__Desulfofervidaceae; g__Desulfofervidus; s__Desulfofervidus auxilii | Guaymas Basin, Gulf of California | [40] |
| JAJSZN000000000 | HotSeep1 FWG170 | | 1777984 | 86.7 | 6.74 | d__Bacteria;p__Desulfobacterota;c__Desulfofervidia;o__Desulfofervidales; f__Desulfofervidaceae;g__Desulfofervidus.s__Desulfofervidus auxilii | Hydrothermal vent Pescadero Basin, Gulf of California | This study |
| JAAXOL000000000 | Seep-SRB1a sp. 1 (str. 013792055) | Syntrophophila gen. nov. sp. 1 | 3174343 | 97.42 | 4.03 | d__Bacteria; p__Desulfobacterota; c__Desulfobacteria; o__Desulfobacterales; f__ETH-SRB1; g__B13-G4; s__B13-G4 sp013792055 | Groundwater from Olkiluoto, Finland | [153] |
| JAJSZT000000000 | Seep-SRB1a sp. 2 (str. CR9063A) | Syntrophophila gen. nov. sp. 2 | 3859851 | 99.35 | 2.15 | d__Bacteria; p__Desulfobacterota; c__Desulfobacteria; o__Desulfobacterales; f__ETH-SRB1; g__B13-G4; s__ | Cold seep, Costa Rica Margin | This study |
| JAJSZU000000000 | Seep-SRB1a sp. 2 (str. CR9063B) | Syntrophophila gen. nov. sp. 2 | 3484925 | 87.56 | 1.94 | d__Bacteria; p__Desulfobacterota; c__Desulfobacteria; o__Desulfobacterales; f__ETH-SRB1; g__B13-G4; s__ | Cold seep, Costa Rica Margin | This study |
| JAJSZV000000000 | Seep-SRB1a sp. 2 (str. CR9063C) | Syntrophophila gen. nov. sp. 2 | 2980801 | 64.05 | 0.00 | d__Bacteria; p__Desulfobacterota; c__Desulfobacteria; o__Desulfobacterales; f__ETH-SRB1; g__B13-G4; s__ | Cold seep, Costa Rica Margin | This study |
| JAAXOL000000000 | Seep-SRB1a sp. 3 (str. 014237195) | Syntrophophila gen. nov. sp. 3 | 3489866 | 98.00 | 1.53 | d__Bacteria; p__Desulfobacterota; c__Desulfobacteria; o__Desulfobacterales; f__ETH-SRB1; g__B13-G4; s__B13-G4 sp014237365 | Cold seep, Gulf of Cadiz | [154] |
| JABZFQ000000000 | Seep-SRB1a sp. 3 (str. 014237365) | Syntrophophila gen. nov. sp. 3 | 3489866 | 98.00 | 1.53 | d__Bacteria; p__Desulfobacterota; c__Desulfobacteria; o__Desulfobacterales; f__ETH-SRB1; g__B13-G4; s__B13-G4 sp014237365 | Cold seep, Gulf of Cadiz | [154] |
| JAJSZO000000000 | Seep-SRB1a sp. 4 (str. FWG171) | Syntrophophila gen. nov. sp. 4 | 2505125 | 77.03 | 0.65 | d__Bacteria;p__Desulfobacterota;c__Desulfobacteria; o__Desulfobacterales; f__ETH-SRB1;g__B13-G4;s__ | Microbial mat, cold seep, Santa Monica Basin, California | This study |
| JAJSZP000000000 | Seep-SRB1a sp. 7 (str. FWG172) | Syntrophophila gen. nov. sp. 7 | 2570901 | 92.1 | 4.73 | d__Bacteria;p__Desulfobacterota;c__Desulfobacteria; o__Desulfobacterales; f__ETH-SRB1;g__B13-G4;s__ | Microbial mat, cold seep, Santa Monica Basin, California | This study |
| JAIORM000000000 | Seep-SRB1a sp. 5 (str. 20073_SRB) | Syntrophophila gen. nov. sp. 5 | 1938412 | 78.03 | 2.60 | d__Bacteria; p__Desulfobacterota; c__Desulfobacteria; o__Desulfobacterales; f__ETH-SRB1; g__B13-G4; s__ | Cold seep, Santa Monica Basin, California, USA | [37] |
| JAIORU000000000 | Seep-SRB1a sp. 5 (str. 20074_SRB) | Syntrophophila gen. nov. sp. 5 | 1678217 | 72.61 | 3.96 | d__Bacteria; p__Desulfobacterota; c__Desulfobacteria; o__Desulfobacterales; f__ETH-SRB1; g__B13-G4; s__ | Cold seep, Santa Monica Basin, California, USA | [37] |
| JAABVG000000000 | Seep-SRB1a sp. 5 (str. S7142MS3) | Syntrophophila gen. nov. sp. 5 | 2766087 | 88.86 | 2.58 | d__Bacteria; p__Desulfobacterota; c__Desulfobacteria; o__Desulfobacterales; f__ETH-SRB1; g__B13-G4; s__ | Cold seep, Santa Monica Basin, California, USA | [25] |
| JAJSZW000000000 | Seep-SRB1a sp. 5 (str. SM7059A) | Syntrophophila gen. nov. sp. 5 | 3861344 | 89.65 | 1.29 | d__Bacteria; p__Desulfobacterota; c__Desulfobacteria; o__Desulfobacterales; f__ETH-SRB1; g__B13-G4; s__ | Cold seep, Santa Monica Basin, California, USA | This study |
| QMMZ00000000 | Seep-SRB1a sp. 6 (str. 003647525) | Syntrophophila gen. nov. sp. 6 | 1690800 | 56.35 | 0.97 | d__Bacteria; p__Desulfobacterota; c__Desulfobacteria; o__Desulfobacterales; f__ETH-SRB1; g__B13-G4; s__B13-G4 sp003647525 | Hydrothermal sediment, Guaymas Basin, Gulf of California | [41] |
| JAFCZG000000000 | Seep-SRB1a sp. 8 (str. AB_03_Bin_172) | Syntrophophila gen. nov. sp. 8 | 5464515 | 52.75 | 9.74 | d__Bacteria; p__Desulfobacterota; c__Desulfobacteria; o__Desulfobacterales; f__ETH-SRB1; g__B13-G4; s__ | Hydrothermal sediment, Guaymas Basin, Gulf of California | [42] |
| JAFDFV000000000 | Seep-SRB1a sp. 9 (str. Meg22_02_Bin_90) | Syntrophophila gen. nov. sp. 9 | 3102898 | 83.52 | 3.87 | d__Bacteria; p__Desulfobacterota; c__Desulfobacteria; o__Desulfobacterales; f__ETH-SRB1; g__B13-G4; s__ | Hydrothermal sediment, Guaymas Basin, Gulf of California | [42] |
| JAFDJY000000000 | Seep-SRB1a sp. 9 (str. Meg22_24_Bin_68) | Syntrophophila gen. nov. sp. 9 | 2156769 | 62.82 | 5.48 | d__Bacteria; p__Desulfobacterota; c__Desulfobacteria; o__Desulfobacterales; f__ETH-SRB1; g__B13-G4; s__ | Hydrothermal sediment, Guaymas Basin, Gulf of California | [42] |
| JAFDKW000000000 | Seep-SRB1a sp. 9 (str.Meg22_46_Bin_236) | Syntrophophila gen. nov. sp. 9 | 2114727 | 57.68 | 1.68 | d__Bacteria; p__Desulfobacterota; c__Desulfobacteria; o__Desulfobacterales; f__ETH-SRB1; g__B13-G4; s__ | Hydrothermal sediment, Guaymas Basin, Gulf of California | [42] |
| JAFDAE000000000 | Desulfobacterales sp. AB_1215_Bin_34* | | 3617733 | 88.46 | 3.59 | d__Bacteria; p__Desulfobacterota; c__Desulfobacteria; o__C00003060; f__C00003106; g__C00003106; s__ | Hydrothermal sediment, Guaymas Basin, Gulf of California | [42] |
| MAXM00000000 | Seep-SRB1g (str. C00003104) | Desulfomellonium gen. nov. sp. 1 | 2209054 | 90.01 | 0.65 | d__Bacteria; p__Desulfobacterota; c__Desulfobacteria; o__C00003060; f__C00003106; g__C00003106; s__C00003106 sp001751015 | Hydrate Ridge, Oregon, USA | [33] |
| MANV00000000 | Seep-SRB1g (str. C00003106) | Desulfomellonium gen. nov. sp. 1 | 2149125 | 90.08 | 0.00 | d__Bacteria; p__Desulfobacterota; c__Desulfobacteria; o__C00003060; f__C00003106; g__C00003106; s__C00003106 sp001751015 | Hydrate Ridge, Oregon, USA | [33] |

*(Continued)*

**Table 1.** (Continued)

| Assembly | Organism | Proposed formal name | Genome size (bp) | Completeness | Contamination | GTDB_classification | Geographic location | Reference |
|---|---|---|---|---|---|---|---|---|
| JAJSZX000000000 | Seep-SRB1g (str. CR10073A) | Desulfomellonium gen. nov. sp. 1 | 3489866 | 99.17 | 0.65 | d__Bacteria; p__Desulfobacterota; c__Desulfobacteria; o__C00003060; f__C00003106; g__C00003106; s__C00003106 sp001751015 | Cold seep, Costa Rica Margin | This study |
| JAJSZY000000000 | Seep-SRB1g (str. CR10073B) | Desulfomellonium gen. nov. sp. 1 | 3824408 | 98.71 | 2.15 | d__Bacteria; p__Desulfobacterota; c__Desulfobacteria; o__C00003060; f__C00003106; g__C00003106; s__C00003106 sp001751015 | Cold seep, Costa Rica Margin | This study |
| JAAABVH00000000 | Seep-SRB1g (str. S7142MS4) | Desulfomellonium gen. nov. sp. 1 | 624626 | 36.28 | 0.16 | d__Bacteria; p__Desulfobacterota; c__Desulfobacteria; o__C00003060; f__C00003106; g__C00003106; s__C00003106 sp001751015 | Cold seep, Santa Monica Basin, California, USA | [25] |
| PQXD00000000 | Seep-SRB2 sp. 1 (str. G37) | Desulfomithrium gen. nov. sp. 1 | 3571848 | 93.83 | 0.00 | d__Bacteria; p__Desulfobacterota; c__Dissulfuribacteria; o__Dissulfuribacterales; f__UBA3076; g__UBA3076; s__UBA3076 sp003194485 | Hydrothermal sediment, Guaymas Basin, Gulf of California | [24] |
| JAFDGK000000000 | Seep-SRB2 sp. 1 (str. Meg22_1012_Bin_255) | Desulfomithrium gen. nov. sp. 1 | 2126229 | 85.79 | 0.6 | d__Bacteria; p__Desulfobacterota; c__Dissulfuribacteria; o__Dissulfuribacterales; f__UBA3076; g__UBA3076; s__UBA3076 sp003194485 | Hydrothermal sediment, Guaymas Basin, Gulf of California | [42] |
| JAFDIH000000000 | Seep-SRB2 sp. 1 (str. Meg22_1214_Bin_80) | Desulfomithrium gen. nov. sp. 1 | 2269757 | 91.15 | 0.6 | d__Bacteria; p__Desulfobacterota; c__Dissulfuribacteria; o__Dissulfuribacterales; f__UBA3076; g__UBA3076; s__UBA3076 sp003194485 | Hydrothermal sediment, Guaymas Basin, Gulf of California | [42] |
| JAFDIM000000000 | Seep-SRB2 sp. 1 (str. Meg22_1416_Bin_176) | Desulfomithrium gen. nov. sp. 1 | 1724888 | 60.53 | 1.98 | d__Bacteria; p__Desulfobacterota; c__Dissulfuribacteria; o__Dissulfuribacterales; f__UBA3076; g__UBA3076; s__UBA3076 sp003194485 | Hydrothermal sediment, Guaymas Basin, Gulf of California | [42] |
| JAFDIU000000000 | Seep-SRB2 sp. 1 (str. Meg22_1618_Bin_165) | Desulfomithrium gen. nov. sp. 1 | 1927125 | 76.83 | 2.68 | d__Bacteria; p__Desulfobacterota; c__Dissulfuribacteria; o__Dissulfuribacterales; f__UBA3076; g__UBA3076; s__UBA3076 sp003194485 | Hydrothermal sediment, Guaymas Basin, Gulf of California | [42] |
| JAJSZQ000000000 | Seep-SRB2 sp. 2 (str. FWG173) | Desulfomithrium gen. nov. sp. 2 | 2393181 | 87.78 | 0.6 | d__Bacteria;p__Desulfobacterota;c__Dissulfuribacteria; o__Dissulfuribacterales; f__UBA3076;g__UBA3076;s__ | Pescadero Basin, Gulf of California | This study |
| JAJSZR000000000 | Seep-SRB2 sp. 2 (str. FWG174) | Desulfomithrium gen. nov. sp. 2 | 2622393 | 93.73 | 0.68 | d__Bacteria;p__Desulfobacterota;c__Dissulfuribacteria; o__Dissulfuribacterales; f__UBA3076;g__UBA3076;s__ | Pescadero Basin, Gulf of California | This study |
| DFBQ00000000 | Seep-SRB2 sp. 3 (str. 002367355) | Desulfomithrium gen. nov. sp. 3 | 2606474 | 94.03 | 1.98 | d__Bacteria; p__Desulfobacterota; c__Dissulfuribacteria; o__Dissulfuribacterales; f__UBA3076; g__UBA3076; s__UBA3076 sp002367355 | Coal Oil Point, Santa Barbara, California, USA | [44] |
| PQXE00000000 | Seep-SRB2 sp. 4 (str. E20) | Desulfomithrium gen. nov. sp. 4 | 2549842 | 94.62 | 1.79 | d__Bacteria; p__Desulfobacterota; c__Dissulfuribacteria; o__Dissulfuribacterales; f__UBA3076; g__UBA3076; s__UBA3076 sp003194495 | Marine sediment, Elba, Italy | [24] |
| QNAY00000000 | Seep-SRB2 sp. 5 (str. 003645605) | Desulfomithrium gen. nov. sp. 5 | 1911994 | 80.95 | 2.13 | d__Bacteria; p__Desulfobacterota; c__Dissulfuribacteria; o__Dissulfuribacterales; f__UBA3076; g__UBA3076; s__UBA3076 sp003645605 | Hydrothermal sediment, Guaymas Basin, Gulf of California | [41] |
| JAFDFS000000000 | Seep-SRB2 sp. 6 (str. Meg22_02_Bin_69) | Desulfomithrium gen. nov. sp. 6 | 1600510 | 70.11 | 2.7 | d__Bacteria;p__Desulfobacterota;c__Dissulfuribacteria; o__Dissulfuribacterales; f__UBA3076;g__UBA3076;s__ | Hydrothermal sediment, Guaymas Basin, Gulf of California | [42] |
| LQBF00000000 | Seep-SRB2 sp. 7 (str. ML8_D) | Desulfomithrium gen. nov. sp. 7 | 3527240 | 99.38 | 12.95 | d__Bacteria;p__Desulfobacterota;c__Dissulfuribacteria; o__Dissulfuribacterales; f__UBA3076;g__UBA3076;s__ | Mahoney Lake, Canada, British Columbia | [42] |
| JAFDGC000000000 | Seep-SRB2 sp. 8 (str. Meg19_1012_Bin_147) | Desulfomithrium gen. nov. sp. 8 | 2934091 | 90.85 | 1.19 | d__Bacteria;p__Desulfobacterota;c__Dissulfuribacteria; o__Dissulfuribacterales; f__UBA3076;g__UBA3076;s__ | Hydrothermal sediment, Guaymas Basin, Gulf of California | [42] |
| JAFDGV000000000 | Seep-SRB2 sp. 8 (str.Meg22_1012_Bin_335) | Desulfomithrium gen. nov. sp. 8 | 2310472 | 95.49 | 1.19 | d__Bacteria;p__Desulfobacterota;c__Dissulfuribacteria; o__Dissulfuribacterales; f__UBA3076;g__UBA3076;s__ | Hydrothermal sediment, Guaymas Basin, Gulf of California | [42] |
| JAFDIE000000000 | Seep-SRB2 sp. 8 (str. Meg22_1214_Bin_60) | Desulfomithrium gen. nov. sp. 8 | 2351303 | 94.32 | 0.73 | d__Bacteria;p__Desulfobacterota;c__Dissulfuribacteria; o__Dissulfuribacterales; f__UBA3076;g__UBA3076;s__ | Hydrothermal sediment, Guaymas Basin, Gulf of California | [42] |
| JAFDIO000000000 | Seep-SRB2 sp. 8 (str. Meg22_1416_Bin_56) | Desulfomithrium gen. nov. sp. 8 | 2557541 | 77.56 | 5.08 | d__Bacteria;p__Desulfobacterota;c__Dissulfuribacteria; o__Dissulfuribacterales; f__UBA3076;g__UBA3076;s__ | Hydrothermal sediment, Guaymas Basin, Gulf of California | [42] |
| JAFDIR000000000 | Seep-SRB2 sp. 8 (str. Meg22_1618_Bin_149) | Desulfomithrium gen. nov. sp. 8 | 2785779 | 95.51 | 3.08 | d__Bacteria;p__Desulfobacterota;c__Dissulfuribacteria; o__Dissulfuribacterales; f__UBA3076;g__UBA3076;s__ | Hydrothermal sediment, Guaymas Basin, Gulf of California | [42] |
| JAFDLI000000000 | Seep-SRB2 sp. 8 (str. Meg22_46_Bin_87) | Desulfomithrium gen. nov. sp. 8 | 2368215 | 91.41 | 0.15 | d__Bacteria;p__Desulfobacterota;c__Dissulfuribacteria; o__Dissulfuribacterales; f__UBA3076;g__UBA3076;s__ | Hydrothermal sediment, Guaymas Basin, Gulf of California | [42] |
| JAFDLJ000000000 | Seep-SRB2 sp. 8 (str. Meg22_810_Bin_10) | Desulfomithrium gen. nov. sp. 8 | 2635042 | 91.64 | 1.92 | d__Bacteria;p__Desulfobacterota;c__Dissulfuribacteria; o__Dissulfuribacterales; f__UBA3076;g__UBA3076;s__ | Hydrothermal sediment, Guaymas Basin, Gulf of California | [42] |

*These species are closely related to the syntrophic partner but, they are not known to be partners of ANME.

ANME, anaerobic methanotrophic archaea; SRB, sulfate-reducing bacteria.

and Seep-SRB1g are often described as Seep-SRB1 [23,39], a historical name that refers to a polyphyletic clade including SRB that are not partners of ANME. In order to make our analysis more accurate, and to aid future classification of syntrophic SRB, we have been careful to differentiate the different Seep-SRB1 clades with curated genomes and representative trees of 16S rRNA and ribosomal proteins (Table 1 and S2 Fig). Each of the 4 syntrophic SRB clades have evolved from taxonomically divergent ancestors with different metabolic capabilities. While

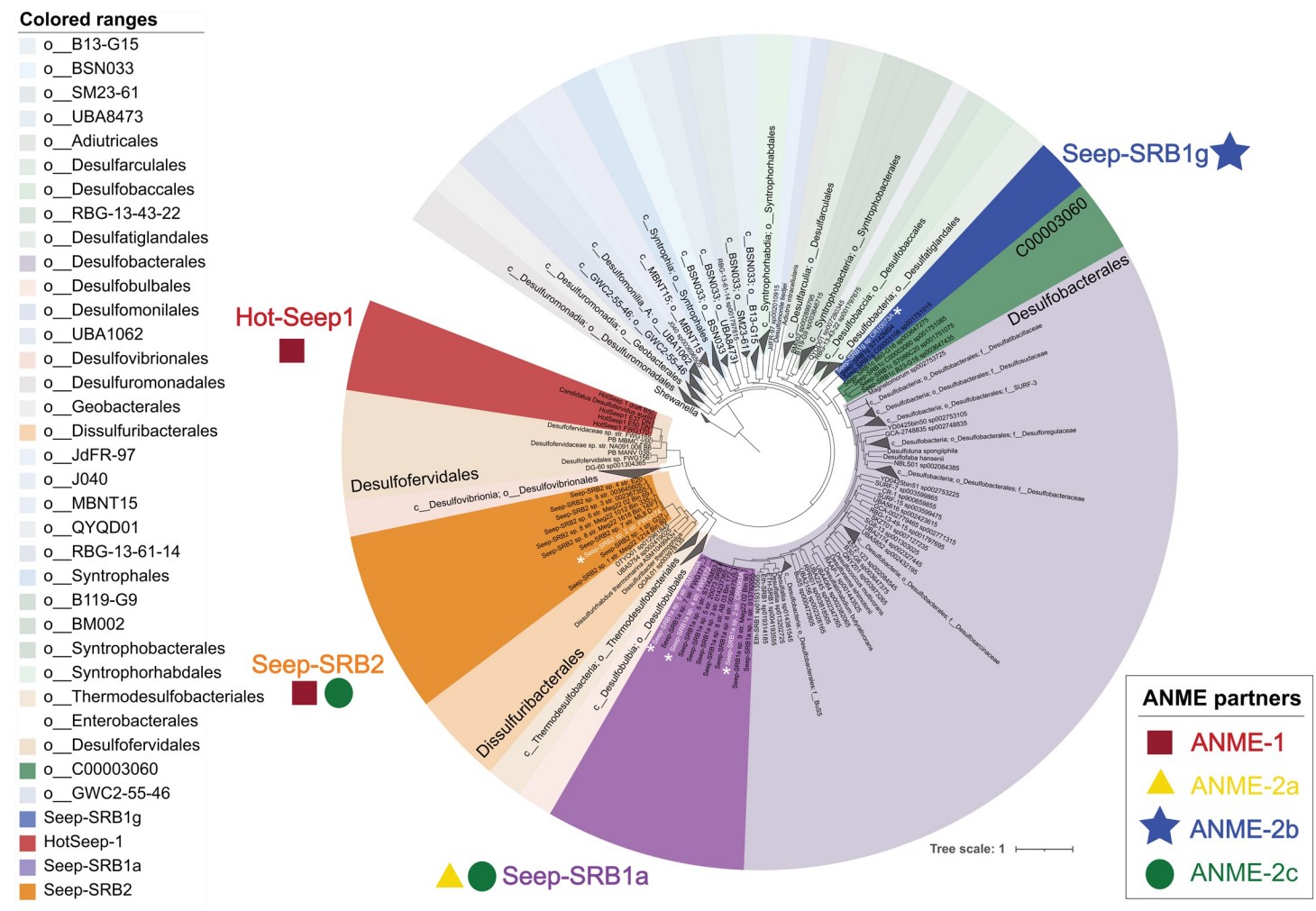

**Fig 1. Taxonomic diversity of syntropic SRB.** A concatenated gene tree of 71 ribosomal proteins from all the Desulfobacterota genomes within the GTDB database release 95 [38] was made using Anvi'o [45]. Genomes from the genus *Shewanella* were used as outgroup. Within this tree, the 4 most common lineages of the syntrophic partners of ANME—Seep-SRB1a, Seep-SRB1g, Seep-SRB2, and Hot-Seep1 are highlighted. While Seep-SRB1a is a genus within the order Desulfobacterales, Seep-SRB1g and Seep-SRB1c together appear to form a closely related order-level taxonomic clade within the class Desulfobacteria. Seep-SRB2 is a genus within the order Dissulfuribacterales while Hot-Seep1 is its own species within the order Desulfofervidales. The proposed type strains are identified on the tree in white with a white asterisk adjacent to the label. The list of genomes used for the generation of concatenated gene tree is listed in S1 Table and the tree is made available as a newick file in S2 Data. ANME, anaerobic methanotrophic archaea; SRB, sulfate-reducing bacteria.

the adaptation to a syntrophic partnership with ANME appears to have been convergently evolved in these clades, their evolutionary trajectories are likely to be different.

Species diversity within each of these clades was inferred by calculating the average nucleotide identity (ANI) (S1 Fig) and 16S rRNA sequence similarity (S2 Table) between different organisms that belong to each clade, using a 95% ANI value and 98.65% similarity in 16S rRNA as cut-offs to delineate different species. Partnership associations, as identified in previous research by our group and others, by FISH [23,24,32], magneto-FISH [53] or FACS sorting [36], and single-aggregate sequencing [37] are depicted in Figs 1 and S1 with further details provided in S1 Text. Briefly, HotSeep-1 has been shown to associate with ANME-1 [24] and other archaea as described above, Seep-SRB2 associate with ANME-2c and ANME-1 [24], SeepSRB1g appears to specifically partner ANME-2b [13] while Seep-SRB1a partners ANME-2a and ANME-2c. All the genomes of Seep-SRB1g in our curated database belong to 1 species-

level clade and thus far, have been shown to partner only ANME-2b [13]. In contrast, there is greater species diversity within the clades that are known to partner more than 1 clade of ANME, Seep-SRB2 and Seep-SRB1a. Whether this diversification is driven by adaptation to partnerships with multiple ANME clades remains to be seen. This pattern is also not consistent with HotSeep-1, a species-level clade that partners multiple archaeal species. A better understanding of the physiological basis for syntrophic partnership formation in each of these clades will provide a framework to understand their unique diversification trajectories.

### Comparative genome analysis of syntrophic SRB

To develop insight into the adaptation of SRB to syntrophic partnerships with ANME, we used a comparative genomics analysis approach to (1) identify the unique features of known syntrophic SRB partners relative to their closest non-syntrophic relatives; and (2) compare the physiological traits that define the diversity within a given clade of syntrophic partner bacteria. For our first objective, we placed the metabolic traits of SRB into the phylogenetic context of the Desulfobacterota phylum, correlating the presence or absence of a physiological trait within the context of genus, family, and order level context of each syntrophic SRB clade. As an example, we demonstrate that the multiheme cytochrome conduit [33] implicated in DIET between ANME and SRB is rare in non-syntrophic Desulfobacterota suggesting that this trait is part of a required adaptation for this syntrophic relationship (Fig 2).

We also investigated the physiological differences between the species of each syntrophic SRB clade. Two of the syntrophic SRB clades, Seep-SRB1g and HotSeep-1 have low species diversity, while the clades Seep-SRB2 and Seep-SRB1a contain multiple species. To better understand the genomic features underlying this diversity, we performed a comparative analysis of species within the Seep-SRB1a and Seep-SRB2 to identify conserved genes across the clade and species-specific genes. A detailed description of the analysis methods is available in Materials and methods and Supporting information (S5 and S6 Figs and S3 and S4 Tables). For this comparative analysis, we primarily focused on pathways that are predicted to be important for the syntrophic interactions between ANME and SRB. In the following section, we describe the pathways within the syntrophic SRB in greater detail and their significance for a syntrophic lifestyle—extracellular electron transfer, inner membrane-bound electron transport chain, electron bifurcation, carbon fixation, nutrient sharing, biofilm formation, cell adhesion, and partner identification. Lastly, we explicitly compare the losses and gains of the genes encoding for the above pathways across the syntrophic SRB and infer the evolutionary trajectory of adaptation towards a syntrophic partnership.

### 1. Respiratory pathways in the 4 syntrophic SRB clades demonstrate significant metabolic flexibility

The respiratory pathways in syntrophic SRB are defined by the necessity of ANME to transfer the electrons derived from methane oxidation to SRB. These electrons are then transferred across the outer membrane to periplasmic electron carriers. These periplasmic electron carriers donate electrons to inner membrane complexes and ultimately, to the core sulfate reduction pathway. Some of the electrons are also used for assimilatory pathways such as carbon fixation. Accordingly, our analysis of the respiratory pathways is split into a description of the pathways for interspecies electron transfer, electron transfer across the inner membrane, and carbon fixation pathways.

**1.1 Multiple pathways exist for interspecies electron transfer between ANME and syntrophic SRB.** The dominant mechanism of interspecies electron transfer between ANME and SRB was proposed to be DIET. This hypothesis is supported by the presence of multiheme

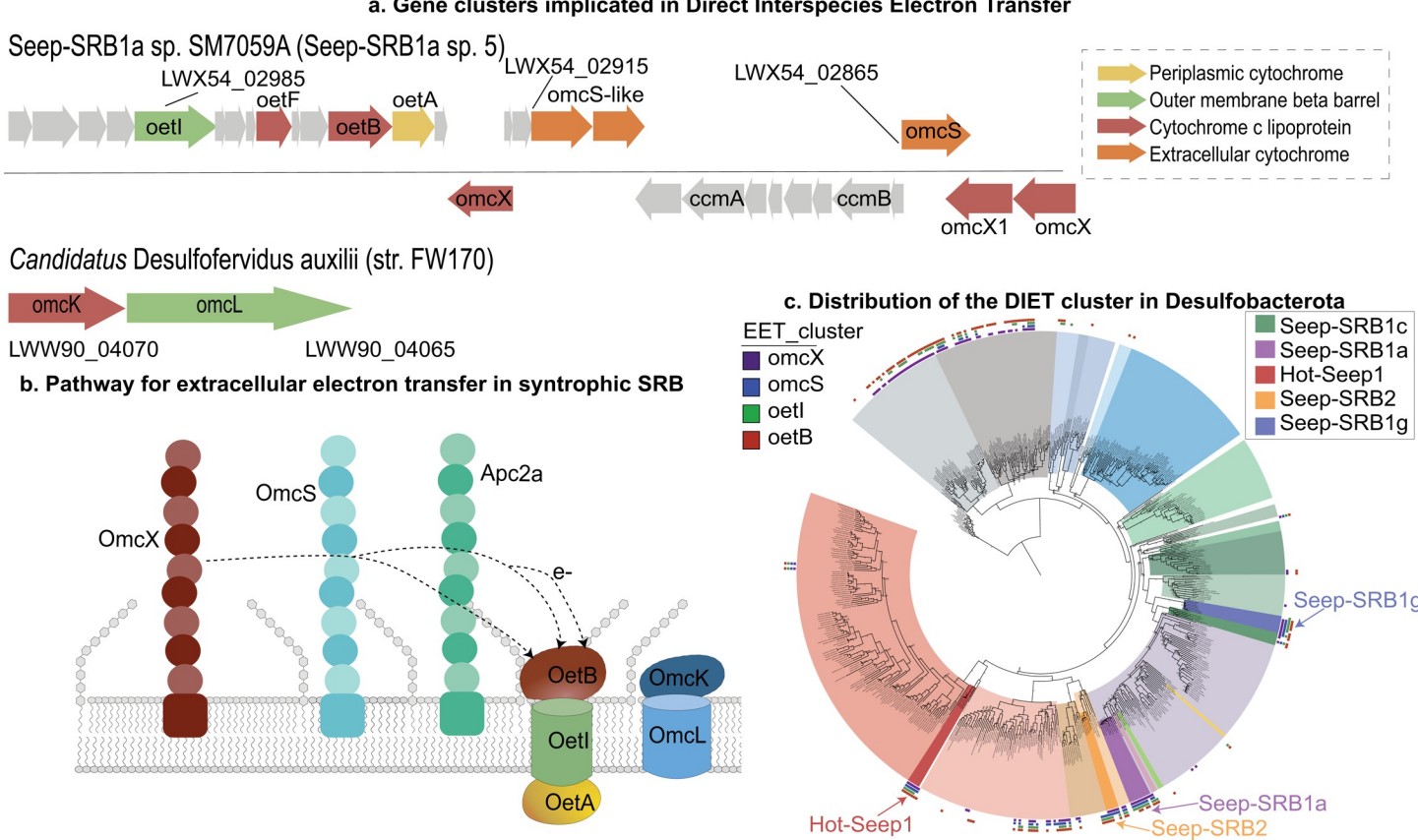

**Fig 2. Gene organization and distribution of the putative cluster implicated in DIET from syntrophic SRB.** (a) The syntenic blocks of genes implicated in DIET including the putative EET conduit OetABI and the operon encoding for OmcKL from HotSeep-1 (*Candidatus* Desulfofervidus auxilii). (b) A model of the putative EET within syntrophic SRB. ANME electrons are likely to be accepted by 1 of 3 putative nanowires formed by multiheme cytochromes homologous to OmcX, OmcS, and a cytochrome we named Apc2a. The electrons from this nanowire would then be transferred to the porin:multiheme cytochrome *c* conduits formed by OetABI or OmcKL and ultimately to different periplasmic cytochromes *c*. (c) The distribution of the putative DIET cluster in the phylum Desulfobacterota is mapped onto a whole genome phylogenetic tree of Desulfobacterota represented in Fig 1 based on the presence of OmcX, OetI, OetB, and OmcS. This cluster is not widely found except in the orders Desulfuromonadales and Geobacterales and the classes Desulfobulbia and Thermodesulfobacteria. ANME, anaerobic methanotrophic archaea; DIET, direct interspecies electron transfer; EET, extracellular electron transfer; SRB, sulfate-reducing bacteria.

cytochromes in genomes of ANME-2a, 2b, and 2c [10], the presence of nanowire-like structures that extend between ANME-1 and its partners Hot-Seep1 [9] and Seep-SRB2 [24], and the presence of hemes in the extracellular space between archaeal and bacterial cells in ANME-SRB aggregates [10,24]. This hypothesis was also supported by the presence of a putative large multiheme cytochrome:porin type conduit, analogous to the conduits in *Geobacter* sp. [54] and other gram-negative bacteria that have been shown to participate in EET [54], in Seep-SRB1g [33], Seep-SRB2 [24], and Hot-Seep-1 [9]. Our analysis of a more comprehensive dataset of syntrophic bacterial genomes confirms the presence of this porin:cytochrome *c* conduit in all the 4 syntrophic bacterial clades studied (S5 Table). Henceforth, we refer to this as the as the (**O**uter-membrane bound **e**xtracellular electron **t**ransfer) or OetI-type conduit. This conduit includes a periplasmic cytochrome *c* (OetA), an outer-membrane porin (OetI), and extracellular facing cytochrome *c* lipoprotein (OetB) (Figs 2B and 3C). The OetI-type conduit was first identified in *G. sulfurreducens* and is expressed when a *Geobacter* mutant of omcB is grown on Fe(III) oxide [55]. The oetABI cassette is found in all 4 syntrophic SRB clades and often includes 2 or 3 other putative extracellular cytochromes *c*, including homologs of OmcX

[33], OmcS (Supplementary alignment MSA1) and a 6-heme cytochrome that we termed apc2a (S5 Table). If they are not found as part of the oet cluster, they could be found elsewhere on the genome, possibly due to genomic rearrangement after acquisition of the cassette (S6 and S7 Tables). The omcX and omcS-like genes in the oet gene cassette are often found in an analogous position to omcS and omcT in G. *sulfurreducens* (Fig 2). Based on the homology of one of the cytochromes to OmcS, which polymerizes to form long and highly conductive filaments that facilitate EET in *Geobacter* [56], we propose that the extracellular cytochromes *c* in this gene cassette perform a similar function, forming filaments that accept electrons from ANME. This is consistent with heme staining of the intercellular space between ANME and SRB, and the observation of filaments that connect the partners [10,24]. This is also consistent with the fact that different extracellular cytochromes are among the most highly expressed proteins in the syntrophic SRB: ANME-1/Seep-SRB2 [24] (OmcX, OmcS-like, and Apc2a), ANME-1/HotSeep-1 [24] (OmcX and OmcS-like), ANME-2c/Seep-SRB2 [24] (OmcX) aggregates and ANME-2a/Seep-SRB1a [25] (OmcX, OmcS-like). The presence of multiple copies of these putative filament-forming proteins in the syntrophic SRB genomes is indicative of their importance to the physiology of syntrophic SRB. The mechanism of electron transfer from extracellular cytochrome filaments to the interior of the cells in *Geobacter* is not well understood. However, a porin:cytochrome *c* conduit is always expressed under the same conditions as a cytochrome *c* containing filament in *Geobacter* (omcS along with extEFG or omcABC under Fe (III) oxide reducing conditions and omcZ along with extABCD during growth on an electrode [57]) and in ANME-SRB consortia (OmcS/OmcX with OetABI or OmcKL). These findings suggest that each cytochrome *c* filament could act in concert with a porin:cytochrome *c* conduit (Fig 2) to transfer electrons from the extracellular space to the periplasm.

While oetABI is conserved in all 4 syntrophic SRB clades, there are 2 other putative porin: cytochrome *c* conduits in syntrophic SRB. A porin (HS1_RS02765) and extracellular cytochrome c (HS1_RS02760) homologous to OmcL and OmcK from G. *sulfurreducens* is found in HotSeep-1 (S6 and S7 Tables) and expressed at a 4-fold higher level than the oetABI conduit [24]. OmcK and OmcL were also up-regulated in G. *sulfurreducens* when it is grown on hematite and magnetite [58]. There is no gene encoding a periplasmic cytochrome *c* adjacent to these genes and this is unusual for previously characterized EET conduits but, given the large number of periplasmic cytochromes in HotSeep-1, it is conceivable that another cytochrome *c* interacts with the OmcL/K homologs. This conduit is also found in Seep-SRB2 sp. 1, 2, 7, and 8 but does not appear to be expressed as highly as the OetABI in the ANME-1/Seep-SRB2 consortia [24]. A different putative conduit including the porin, extracellular, and periplasmic cytochromes *c* is present in the Seep-SRB1g genomes (LWX52_07950- LWX52_07960) (S6 and S7 Tables). This conduit does not have identifiable homologs in *Geobacter*. The presence of multiple porin:cytochrome *c* conduits in the syntrophic partners suggests some flexibility in use of electron donors, possibly from different syntrophic partners. For HotSeep-1, this observation is consistent with its ability to form partnerships with both methane and other alkane-oxidizing archaea [28]. The role of the second conduit is less clear in Seep-SRB1g which to date has only been shown to partner with ANME-2b [13]. Future investigation of the multiple syntrophic SRB EET pathways and the potential respiratory flexibility it affords to their partner archaea using transcriptomics, proteomics and possibly heterologous expression methods will further expand our understanding of electron transfer in these diverse consortia.

While DIET is believed to be the dominant mechanism of syntrophic coupling between the ANME and SRB partners, the potential to use diffusible intermediates such as formate and hydrogen exists in some genomes of syntrophic SRB. Hydrogenases are present in HotSeep-1, which can grow without ANME using hydrogen as an electron donor [32]. We also identified periplasmic hydrogenases in Seep-SRB1a sp. 1, 5, and 8 (S7 Table) that suggest that these

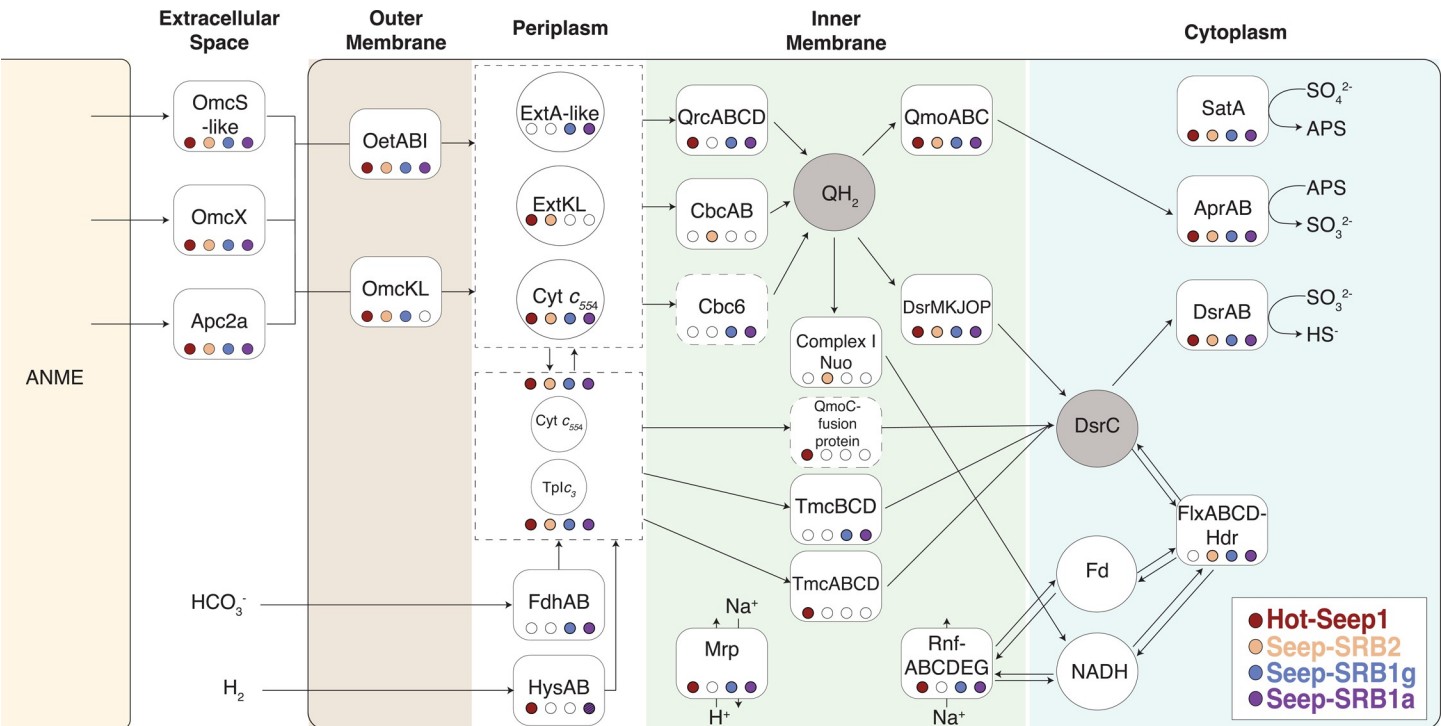

**Fig 3. Summary of the different electron transport chains in syntrophic SRB.** The various respiratory proteins essential for the electron transport chain within the syntrophic SRB are identified and marked within their predicted cellular compartments. Filled circles indicate their presence in each of the 4 syntrophic sulfate-reducing bacterial clades, HotSeep-1 (red), Seep-SRB2 (orange), Seep-SRB1g (blue), and Seep-SRB1a (purple). The 2 typical acceptors of electrons transferred across the inner membrane, quinols ($QH_2$) and DsrC, are indicated in shaded circles. These are the 2 nodes which much of the respiratory flexibility of the syntrophic SRB revolves around. SRB, sulfate-reducing bacteria.

organisms could use hydrogen as an electron donor. However, in Seep-SRB1a, these hydrogenases are expressed at low levels (less than a 20th of the levels of DsrB) in the ANME-2a/Seep-SRB1a consortia [25]. Further, previous experiments showed that the addition of hydrogen to ANME-2/SRB consortia did not inhibit anaerobic oxidation of methane suggesting that hydrogen is not the predominant agent of electron transfer between ANME and SRB [35,59]. Perhaps, hydrogenases are used by Seep-SRB1a to scavenge small amounts of hydrogen from the environment. While membrane bound and periplasmic hydrogenases are present in non-syntrophic Seep-SRB1c (S7 Table), no hydrogenases are found in the syntrophic relative of Seep-SRB1c and ANME partner, Seep-SRB1g. Similarly, periplasmic hydrogenases are present in Dissulfuribacteriales and absent in Seep-SRB2 (one exception in 18 genomes), suggesting that in both these partners, the loss of periplasmic hydrogenases is part of the adaptation to their syntrophic partnership with ANME. We also identified periplasmic formate dehydrogenases in Seep-SRB1g and Seep-SRB1a sp. 2, 3, 8, 9 (S7 Table). The periplasmic formate dehydrogenase from Seep-SRB1g is expressed in the environmental proteome at Santa Monica Mounds [33], but no transcripts from the formate dehydrogenases of Seep-SRB1a were recovered in the ANME-2a/Seep-SRB1a incubations [25]. It is possible that these syntrophic SRB scavenge formate from the environment. Alternatively, a recent paper found a hybrid of electron transfer by DIET and by diffusible intermediates (mediated interspecies electron transfer or MIET) to be energetically favorable [60]. In this model, the bulk of electrons would still be transferred by DIET, but up to 10% of electrons could be shared by MIET via formate [60], an intermediate suggested in earlier studies [35,59]. This might be possible in ANME/SRB consortia with

HotSeep-1, some species of Seep-SRB1a and Seep-SRB1g, but not in ANME/Seep-SRB2 consortia. The absence of periplasmic formate dehydrogenases and hydrogenases in Seep-SRB2 as previously observed [24] is also true in our expanded dataset. If a diffusive intermediate should play a role in mediating electron transfer between ANME-2c or ANME-1 and Seep-SRB2, it is not likely to be formate or hydrogen.

**1.2 Pathways for electron transfer across the inner membrane vary in different syntrophic SRB clades.** Multiheme cytochromes *c* in SRB are known to mediate diverse modes of electron transfer from different electron donors to a conserved sulfate reduction pathway [58]. There is significant variety in the number and types of cytochromes *c* present in SRB from the phylum Desulfobacterota [61] and an even greater number of large cytochromes is present in syntrophic SRB [24,33]. To explore the potential for different routes of electron transfer, we performed an analysis of all cytochromes *c* containing 4 hemes or more from the genomes of syntrophic SRB (see Materials and methods) and identified at least 27 different types of cytochromes *c*. We split these cytochromes *c* into those predicted to be involved in EET, those that act as periplasmic electron carriers and those that are components of protein complexes involved in electron transfer across the inner membrane (S6 Table). Conserved across the syntrophic SRB partners of ANME were the cytochromes forming the core components of the EET pathway—OetA, OetB, OmcX and OmcS-like and Apc2a extracellular cytochromes, and 2 periplasmic cytochromes of the types, TpI$c_3$ [62] and cytochrome $c_{554}$ [61,63]. Beyond the conserved periplasmic cytochromes *c*, TpI$c_3$ and cytochrome $c_{554}$, there are also cytochromes binding 7–8 hemes that are unique to different SRB clades (S6 Table). These include a homolog of ExtKL [64] from *G. sulfurreducens* that is highly expressed in Seep-SRB2 spp. 1 and 4 during growth in a syntrophic partnership with ANME [24] and a homolog of ExtA from *G. sulfurreducens* [54] protein expressed in the ANME-2a/Seep-SRB1a consortia [25]. Previous research has suggested that the tetraheme cytochromes *c* are not selective as electron carriers and play a role in transferring electrons to multiple different protein complexes [65]. It is possible that these larger 7–8 heme binding cytochromes *c* have a more specific binding partner. Both the ExtKL and ExtA-like proteins are very similar (over 45% sequence similarity) to their homologs in Desulfuromonadales. Since the OetI-type conduit is also likely transferred from this order, they might act as binding partners to the OetI-type conduit transferring electrons to the periplasmic cytochromes *c*.

In SRB, the electrons from periplasmic electron donors (reduced by DIET or MIET) are delivered through inner membrane bound complexes to quinones or directly to the heterodisulfide DsrC in the cytoplasm via transmembrane electron transfer [61] (Fig 3). The electrons from quinones or DsrC are ultimately used for the sulfate reduction pathway (including SatA, AprAB, and DsrAB) [24,33,61]. Two conserved protein complexes are always found along with this pathway—the Qmo complexes transfers electrons from reduced quinones to AprAB and the DsrMKJOP complexes transfers electrons from quinones to DsrC and through DsrC to DsrAB. Since both these complexes use electrons from reduced quinones, the source of reduced quinones in the inner membrane is critical to different sulfate respiration pathways. The quinol reducing complexes and complexes that reduce DsrC provide respiratory flexibility to SRB. We also note here that the reduction of AprAB coupled to the oxidation of menaquinone is expected to be endergonic. There is a proposal that QmoABC might function through flavin-based electron confurcation (FBEC), using electrons from reduced quinones and a second electron donor such as ferredoxin to reduced AprAB [66]. Since, it is not clear what the electron donor is likely to be, we do not explicitly consider this reaction in our analysis. A summary of all the putative complexes that are involved in the electron transport chains of the 4 syntrophic SRB is visualized in Fig 3 to detail how electron transport pathways vary among the clades. A more detailed list of complexes present is found in S6–S8 Tables. The respiratory

pathways in HotSeep-1, Seep-SRB1a, and Seep-SRB1g are broadly similar in structure and are predicted to use the Qrc complex to transfer periplasmic electrons to the quinone pool and both DsrMKJOP and Tmc to reduce cytoplasmic DsrC. Their pathways are analogous to the respiratory pathways in *Desulfovibrio alaskensis* [62,67]. Curiously, the Tmc complex in Seep-SRB1a and Seep-SRB1g are divergent from Tmc in non-syntrophic SRB (S8 Fig) and TmcA is absent in the operons encoding for Tmc. This absence suggests that Tmc has been adapted to use a different electron donor than in *Desulfovibrio vulgaris* [67] and is consistent with the fact that the electron donor for Seep-SRB1a and Seep-SRB1g is not hydrogen or formate but, electrons from anaerobic methanotrophic archaea. It is not clear why Tmc is more divergent than DsrMKJOP in syntrophic SRB compared to their evolutionary neighbors since they are both likely important for DsrC reduction and are equally well expressed under methane-oxidizing conditions [24,25]. Qrc is known to be important for energy conservation in this respiratory pathway. Protons are translocated by Qrc from the cytoplasmic side to the periplasmic active side. This movement of charges across the membrane leads to the generation of proton motive force (pmf) that can be utilized by ATP synthase to generate ATP [68]. In Seep-SRB1a, Seep-SRB1g, and HotSeep-1, Qrc likely acts with the conserved DsrMKJOP and QmoABC to generate pmf. While purified Dsr and Qmo have not yet been shown to be electrogenic, it is expected that DsrMKJOP [69] and QmoABC [69,70] might be able to generate pmf by charge translocation.

In Seep-SRB2, Qrc is absent and we hypothesize that CbcBA, a protein complex that appears to be horizontally transferred from the Desulfuromonadales (Figs 3 and 4), mediates electron transfer between periplasmic cytochromes *c* and quinones [71]. This is supported by the fact that CbcBA is highly expressed during AOM between ANME-1/Seep-SRB2 and ANME-2c/Seep-SRB2 [24]. In *Geobacter sulfurreducens*, which also does not have Qrc this cytoplasmic membrane-bound oxidoreductase is expressed during growth on Fe(III) at low potential and is important for iron reduction and growth on electrodes at redox potentials less than −0.21 mV [71]. During AOM, the CbcBA protein in Seep-SRB2 is predicted to run in the reverse direction, reducing quinols using electrons from DIET electrons supplied by ANME as opposed to functioning in the metal reducing direction. While the reversibility of this complex has not been biochemically established, the high levels of expression of this complex suggest that this is likely functional in the electron transport chain of Seep-SRB2. It is not clear what the likely site of energetic coupling is within the Seep-SRB2 respiratory chain. In the absence of the Qrc complex, the most likely mechanism for energetic coupling might exist through the action of a Q-loop mechanism [69]. In this mechanism, energy is conserved by the combined action of 2 protein complexes that reduce and oxidize quinols, leading to the uptake and release of protons on opposite sides of the cytoplasmic membrane. The Q-loop mechanism in Seep-SRB2 would likely involve CbcBA and a quinol oxidizing complex such as Qmo.

In addition to the most likely pathways of electron transfer in the syntrophic SRB, as established using transcriptomic data on ANME/SRB partnership [24,25] (Figs 2 and 3), other inner membrane complexes exist in these genomes that may provide additional respiratory flexibility. HotSeep-1 genomes contain a complex that involves an HdrA subunit and a protein that also binds hemes *c* and contains a CCG domain similar to that found in HdrB and TmcB predicted to interact with DsrC [70,74]. This complex a putative cytochrome *c* oxidoreductase containing a CCG domain, would likely transfer electrons from cytochromes *c* to the DsrC (AMM42179.1-AMM42180.1) or perhaps to a ferredoxin. The presence of HdrA might indicate a role in electron bifurcation by this complex. It is highly expressed during methane oxidation conditions in the ANME-1/HotSeep-1 consortia to a fifth of the level of the Tmc complex that would play a similar role in electron transfer [24]. In some Seep-SRB1a and Seep-SRB1g genomes, there is a homolog of Cbc6 (LWX51_14670- LWX51_14685) identified

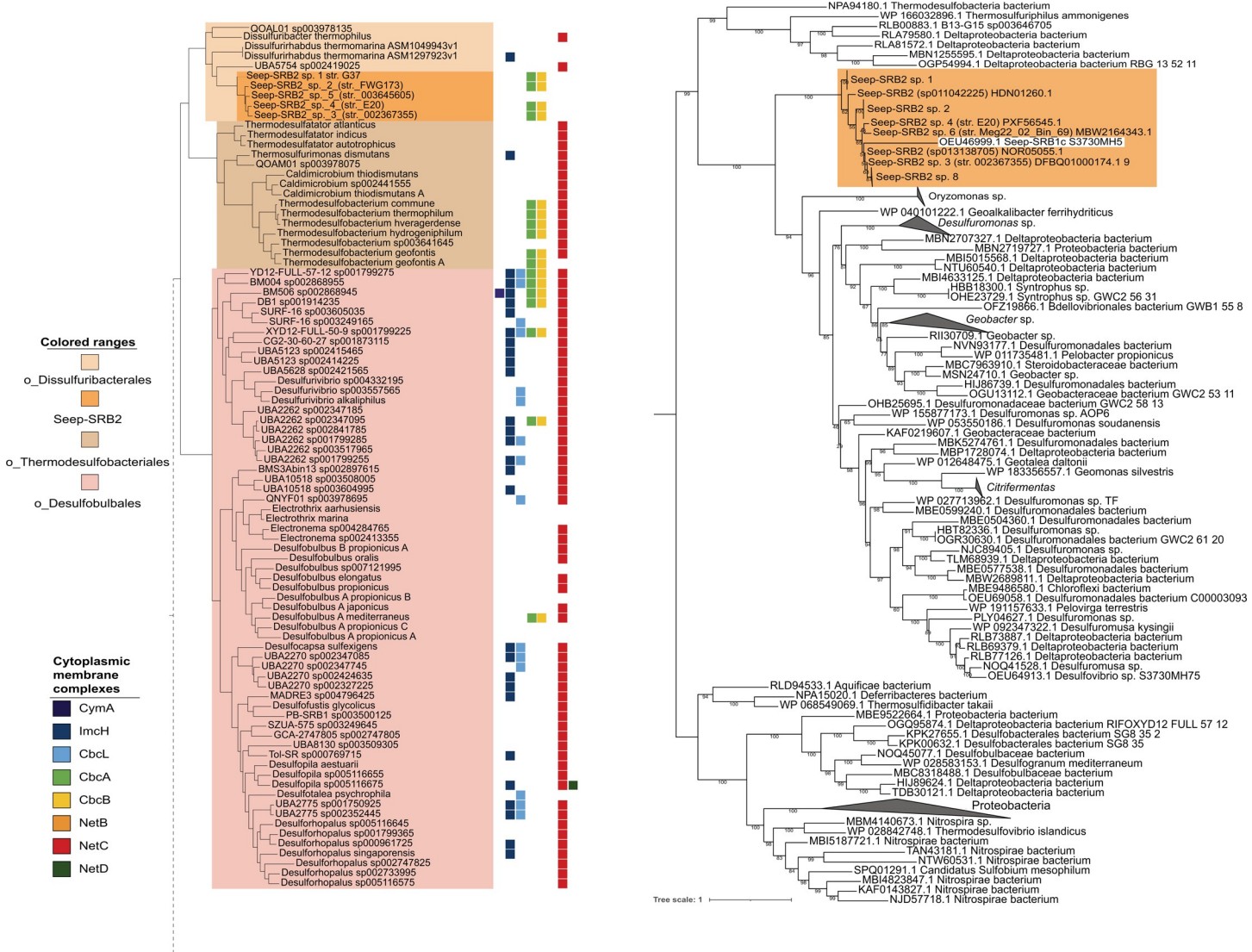

**Fig 4. _cbcBA_ as an example of horizontal gene transfer events into Seep-SRB2.** We demonstrate one example of an important gene transfer event involving a metabolic gene. The function of cbcBA is essential for the central respiratory pathway in Seep-SRB2 and this gene was acquired by horizontal gene transfer. (A) The presence of CymA, CbcL, CbcAB, and NetBCD, commonly used electron donors to the EET conduits in _Shewanella_ and _Geobacter_ are mapped on to the classes Thermodesulfobacteria and Desulfobulbia. (B) CbcB protein sequences were aligned using MUSCLE [72] and then a phylogenetic tree was inferred using IQ-Tree2 [73]. The CbcB sequences from Seep-SRB2 are highlighted in orange. The phylogenetic tree is available in newick format in S2 Data. EET, extracellular electron transfer; SRB, sulfate-reducing bacteria.

in _Geobacter_ [75] and implicated in electron transfer from periplasmic cytochromes _c_ to the quinol pool. A NapC/NirT homolog [76] was conserved in Seep-SRB1g (OEU53943.1-OEU53944.1) and some Seep-SRB2, and another conserved complex that includes a cytochrome _c_ and ruberythrin (AMM39991.1-AMM39993.1) is present in Seep-SRB1g and HotSeep-1. Further research is needed to test whether there are conditions under which these complexes are expressed. Our analysis indicates some degree of respiratory specialization in the syntrophic SRB genomes such as the loss of hydrogenases in Seep-SRB1g and Seep-SRB2 compared to their nearest evolutionary neighbors, suggesting an adaptation towards a partnership with ANME. However, considerable respiratory flexibility still exists within the genomes of these syntrophic partners as is suggested by the presence of the formate

dehydrogenases in Seep-SRB1g and Seep-SRB1a, multiple EET conduits in HotSeep-1 and Seep-SRB2 and multiple inner membrane complexes in Seep-SRB1a and HotSeep-1.

**1.3 Cytoplasmic redox reactions, electron bifurcation, and carbon fixation pathways.** The electron transport chain outlined above would transfer electrons from periplasmic cytochromes $c$ to the cytoplasmic electron carrier DsrC or directly to the sulfate reduction pathway. However, the electron donors for carbon or nitrogen fixation are typically NADH, NADPH, or ferredoxin [77]. The reduction of NADH, NADPH, or ferredoxin could happen through the transfer of electrons from DsrC [78] or through the interconversion of electrons between cytoplasmic electron carriers through the dissipation of pmf or through the action of electron bifurcating complexes [79]. The transfer of electrons from DsrC to these reductants likely happens through the action of protein complexes like Flx-Hdr that can oxidize 2 molecules of NADH to reduce 1 molecule of ferredoxin and 1 molecule of DsrC. Electrons from NADH and ferredoxin can also be exchanged with the dissipation or generation of sodium motive force using membrane-bound Rnf and Mrp [79–81] in Seep-SRB1a, Seep-SRB1g, and HotSeep1. In marine environments, the naturally occurring sodium gradient can be used to generate ferredoxin from NADH or vice versa using the Rnf complex, while the $Na^+/H^+$ antiporter, Mrp, can transport $Na^+$ or $H^+$ in response to the action of Rnf [82]. The ferredoxin generated from this process can then be used for assimilatory pathways. In Seep-SRB2, which does not contain Rnf or Mrp (S9 Fig), the NADH needed for carbon fixation is likely obtained through the oxidation of quinol by complex I and the dissipation of pmf [83]. In addition to Complex I or Rnf and Mrp, there are additional cytoplasmic protein complexes that can recycle reducing equivalents between DsrC, ferredoxin, and NADH such as the electron bifurcating Flx-Hdr [70,78]. Several putative oxidoreductase complexes in the syntrophic SRB genomes are compiled in S7 Table and S10 and S11 Figs.

Syntrophic sulfate-reducing members of the Seep-SRB1a, Seep-SRB1g, and Seep-SRB2 have been shown to fix carbon using the Wood–Ljungdahl pathway, while organisms of the clade HotSeep-1 partnering with ANME-1 are predicted to fix carbon using the reductive tricarboxylic acid cycle (rTCA) [24,33,77]. Analysis of gene synteny for a number of Seep-SRB1a, Seep-SRB1g, and Seep-SRB2 MAGs uncovered a number of heterodisulfide (HdrA) subunits and HdrABC adjacent to enzymes involved in the Wood–Ljungdahl pathway (S10 Fig). These subunits are typically implicated in flavin-based electron bifurcating reactions utilizing ferredoxins or heterodisulfides and NADH [79]. Specifically, Seep-SRB1g has an HdrABC adjacent to metF that is predicted to encode for a putative metF-HdrABC, performing the reduction of methylene tetrahydrofolate coupled to the endergonic reduction of ferredoxin to NADH, the same reaction as the bifurcating metFV-HdrABC described below. In Seep-SRB1g, there are also 2 copies of HdrABC next to each other whose function requires further analysis (S10 Fig). These complexes are absent in the related group Seep-SRB1c, a lineage which has not yet been found in physical association with ANME (S10 Fig). The presence of electron bifurcation machinery in the carbon fixation pathways within several syntrophic SRB lineages, suggests that they are optimized to conserve energy (S10 Fig). This is reminiscent of the MetFV-HdrABC in the acetogen *Moorella thermoacetica* [79] in which the NADH-dependent methylene tetrahydrofolate reduction within the central metabolic pathway is coupled to the endergonic reduction of ferredoxin by NADH, allowing for the recycling of reducing equivalents. Members of the Seep-SRB1g also have a formate dehydrogenase (fdhF2) subunit adjacent to nfnB, the bifurcating subunit of NfnAB, which performs the NADPH-dependent reduction of ferredoxin (S11 Fig). This complex is predicted to function as an additional bifurcating enzyme that would allow for the recycling of NADPH electrons. In addition, HotSeep-1, Seep-SRB2, and Seep-SRB1g appear to have homologs of electron transfer flavoproteins, etfAB, that are expected to be electron bifurcating. These homologs of etfAB cluster with the previously

identified bifurcating etfAB and possess the same sequence motif that was previously shown to correlate with the electron bifurcating etfAB [84] (S7 Table). While the capability of electron bifurcation by these enzyme complexes needs to be biochemically confirmed, the possibility of a high number of bifurcating complexes, especially those connected to the carbon fixation pathway, in the genomes of syntrophic SRB partners of ANME is compelling. It could be argued that this is a natural adaptation to growth in very low-energy environments or to low-energy metabolism. In fact, some of these complexes are present in non-syntrophic bacteria of the order Desulfofervidales and genus Eth-SRB1. These adaptations could provide an additional energetic benefit for the syntrophic lifestyle, itself an adaptation to low-energy environments.

## 2. Cobalamin auxotrophy and nutrient sharing in syntrophic SRB

Research on the AOM symbiosis has focused heavily on the nature of the syntrophic intermediates shared between ANME and SRB [9,10,12,85,86]. We currently have an incomplete understanding of the scope of other potential metabolic interdependencies within this long-standing symbiosis. Prior experimental research has demonstrated the potential for nitrogen fixation and exchange in AOM consortia under certain environmental circumstances [11,13,87,88], and in other energy limited anaerobic syntrophies between bacteria and archaea, amino acid auxotrophies are common [89–91]. Comparative analysis of MAGs from several lineages of ANME [14] as well as a subset of syntrophic sulfate-reducing bacterial partners [33] lacked evidence for specific loss of pathways used in amino acid synthesis, and our expanded analysis of SRB here is consistent with these earlier studies. Interestingly, comparative analysis of specific pairings of ANME and their SRB partners revealed the possibility for cobalamin dependency and exchange. Cobamides, also known as the Vitamin B12-type family of cofactors, are critical for many central metabolic pathways [92]. Mechanisms for complete or partial cobamide uptake and remodeling by microorganisms found in diverse environments are common [92]. The importance of exchange of cobamide between gut bacteria and between bacteria and eukaryotes has been demonstrated [93,94]. In methanotrophic ANME-SRB partnerships, ANME are dependent on cobalamin as a cofactor in their central metabolic pathway and biosynthetic pathways, while Seep-SRB2, Seep-SRB1a, Seep-SRB1g also have essential cobalamin-dependent enzymes including ribonucleotide reductase, methionine synthase, and acetyl-CoA synthase (S8 Table). This is in contrast with the HotSeep-1 clade, which appears to have fewer cobalamin requiring enzymes and may not have an obligate dependence on vitamin B12. However, HotSeep-1 do possess homologs of BtuBCDF and CobT/CobU, genes that are used in cobamide salvage and remodeling [95] (S8 Table). An absence of cobalamin biosynthesis in either ANME or these 3 clades of syntrophic SRB would thus necessarily lead to a metabolic dependence on either the partner or external sources of cobalamin in the environment. We observed such a predicted metabolic dependence for Seep-SRB1a within the species Seep-SRB1a sp. 1 ($n = 1$ genomes), Seep-SRB1a sp. 5 ($n = 4$ genomes), Seep-SRB1a sp. 3 ($n = 2$ genomes), Seep-SRB1a sp. 7 ($n = 1$), and Seep-SRB1a sp. 8 ($n = 1$). All these genomes are missing the anaerobic corrin ring biosynthesis pathway but, some do retain genes involved in lower ligand synthesis (BzaAB) [96] (Fig 5). Additionally, recent metatranscriptomic data from an AOM incubation dominated by ANME-2a/Seep-SRB1a associated with Seep-SRB1a sp. 5 (str. SM7059A) that is missing the cobalamin biosynthesis pathways confirmed active expression of cobalamin-dependent pathways in the Seep-SRB1a including ribonucleotide reductase and acetyl-coA synthase AcsD [25], suggesting that these syntrophs must acquire cobalamin from their ANME partner or the environment.

Interestingly, the predicted cobalamin auxotrophy is not a uniform trait within the Seep SRB1a lineage, with cobamide biosynthesis genes present in the genomes of species Seep-SRB1a sp. 2 ($n = 3$), Seep-SRB1a sp. 4 ($n = 1$), and Seep-SRB1a sp. 9 ($n = 3$) (Fig 5). Of the 5 species missing cobalamin biosynthesis pathways, 2 are verified ANME-2a partners. Of the 4 species containing cobalamin biosynthesis pathways, one is a verified ANME-2c partner and one was sequenced from a microbial mat that contains ANME-2c and ANME-2a (Fig 5). These patterns suggest that the Seep-SRB1a partners of ANME-2a developed a nutritional auxotrophy that is specific to this partnership. Future experimental work will assist with testing this predicted vitamin dependency among the ANME-2a and Seep-SRB1a and other ANME-SRB partner pairings.

The ability to fix nitrogen is found in bacteria and archaea but is relatively rare among them [97]. Fixed nitrogen availability can impact the productivity of a given ecosystem. Members of the ANME-2 archaea have been demonstrated to fix nitrogen in consortia [11,12,88] and may serve as a source of fixed nitrogen for methane-based communities in deep-sea seeps [88]. We recently demonstrated that within the ANME-2b/Seep-SRB1g partnership, Seep-SRB1g bacteria can also fix nitrogen [13]. A comparison of the nitrogen fixation ability across ANME and SRB (Fig 5) shows that this function is present in the genome representatives of diverse ANME and also conserved in some syntrophic bacterial partners (Seep-SRB1a and Seep-SRB1g). In the Seep-SRB1a lineage, the nitrogenase operon is retained in both ANME-2a and ANME-2c partners, contrasting the pattern observed with cobalamin synthesis. Interesting, the potential to fix nitrogen occurs in species of Seep-SRB2 that come from psychrophilic deep-sea environments (Seep-SRB2 sp. 4 and Seep-SRB2 sp. 3), while earlier branching clades of Seep-SRB2 adapted to hotter environments (Seep-SRB2 sp. 1 and 2) lack nitrogenases, hinting at potential ecophysiological adaptation to temperature (Fig 5). While the ability to fix nitrogen is retained in several clades of syntrophic SRB, previous stable isotope labeling experiments have shown that ANME is the dominant nitrogen fixing partner [11,13,88]. Yet, the potential to fix nitrogen is retained in Seep-SRB1a and Seep-SRB1g members (Fig 5), and in some cases, have been directly linked to $N_2$ fixation in the case of Seep SRB1g [13] or indirectly suggested from the recovery of nifH transcripts belonging to Seep-SRB1a and Seep-SRB1g in seep sediments [12]. These observations indicate that nitrogen sharing dynamics between ANME and SRB is likely more complicated than we have thus far observed and may correspond to differences in environment, or perhaps to specific partnership interactions that require assessment at greater taxonomic resolution.

## 3. Pathways related to biofilm formation and intercellular communication

ANME and SRB form multicellular aggregates in which they are spatially organized in distinct and recognizable ways [98]. ANME-2a/2b/2c and ANME-3 are known to form tight aggregates with their bacterial partners [10,25,35,98]. Some members of ANME-1 have been observed in tightly packed consortia with SRB [24], while others some form more loose associations [19,26,99,100]. In these consortia, archaeal and bacterial cells are often enmeshed in an extracellular polymeric substance [19,20,101]. In large carbonate associated mats of ANME-2c and ANME-1 and SRB from the Black Sea, extractions of exopolymers consisted of 10% neutral sugars, 27% protein, and 2.3% uronic acids [20]. This composition is consistent with the roles played by mixed protein and extracellular polysaccharide networks shown to be important for the formation of conductive biofilms in *Geobacter sulfurreducens* [102], the formation of multicellular fruiting bodies from *Myxococcus xanthus* [103–105], and the formation of single-species [106] and polymicrobial biofilms [107]. Important and conserved features across these biofilms are structural components made up of polysaccharides, cellular extensions such as

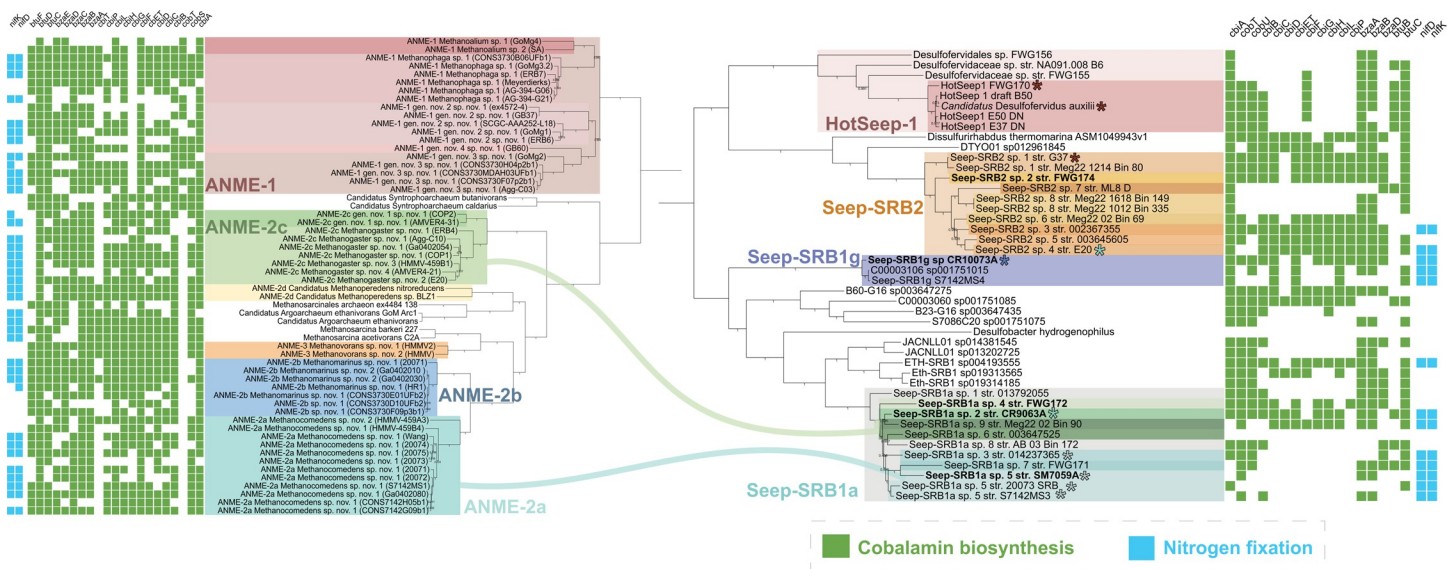

**Fig 5. The loss of cobalamin biosynthesis genes in the Seep-SRB1a partners of ANME-2a.** On the right, a phylogenetic tree of concatenated ribosomal proteins from all the genomes of syntrophic SRB clades—Hot-Seep1, Seep-SRB2, Seep-SRB1g, and Seep-SRB1a and related clades, Seep-SRB1c and Eth-SRB1 was made using Anvi'o [45] and made available in S2 Data. On the left, a similar concatenated protein tree (available in S2 Data) was made for ANME genomes highlighting the clades from ANME-1, ANME-2c, ANME-2b, and ANME-2a. Lines in green and light teal are used to depict the partnerships between ANME-2c and verified species of Seep-SRB1a, and ANME-2a and verified species of Seep-SRB1a, respectively. ANME-2c genomes are not separated into those belonging to partners of Seep-SRB2 and Seep-SRB1a. The presence of genes involved in cobalamin biosynthesis and nitrogen fixation are marked in light green and light blue, respectively. The proposed type strains are bolded. ANME, anaerobic methanotrophic archaea; SRB, sulfate-reducing bacteria.

type IV pili and matrix-binding proteins such as fibronectin-containing domains [108]. Functional components of the biofilm matrix such as virulence factors in pathogens [109] and EET components [102] are variable and depend on the lifestyle of the microorganism. Guided by the molecular understanding of mechanisms and physiological adaptation to microbial growth in biofilms, we examined the genomic evidence for similar adaptations in the syntrophic SRB in consortia with ANME, focusing on structural and functional components of biofilms as well as proteins implicated in partner identification (Fig 6).

**3.1 Multiple polysaccharide biosynthesis pathways are found in syntrophic SRB.** Our analysis of syntrophic SRB genomes showed the presence of multiple putative polysaccharide biosynthesis pathways in different SRB lineages including secreted extracellular polysaccharide biosynthesis pathways and capsular polysaccharide biosynthesis pathways (S9 Table). In particular, homologs of the pel biosynthesis pathway (PelA, PelE, PelF, and PelG), first identified in *Pseudomonas aeruginosa* [110,111] were present in almost all Seep-SRB1g and Seep-SRB1a genomes (S12 and S13 Figs). These homologs are part of a conserved operon in these genomes which includes a transmembrane protein that could perform the same function as PelD, which along with PelE, PelF, and PelG forms the synthase component of the biosynthetic pathway and enables transport of the polysaccharide pel across the inner membrane [111]. Metatranscriptomic data confirms this operon is expressed and was significantly down-regulated when methane-oxidation by ANME-2a was decoupled from its syntrophic Seep SRB1a partner with the addition of AQDS [25]. This biosynthesis pathway is absent in the nearest evolutionary neighbors of Seep-SRB1a and Seep-SRB1g, Eth-SRB1 and Seep-SRB1c, respectively, suggesting that the presence of the pel operon could serve as a better genomic marker for syntrophic interaction with ANME-2a, ANME-2b, and ANME-2c than the presence of the oetI-type conduit. The pel operon was also detected in one of the Seep-SRB2 genomes but is not conserved across this clade. In Seep-SRB2 clades, multiple capsular polysaccharide biosynthesis pathways

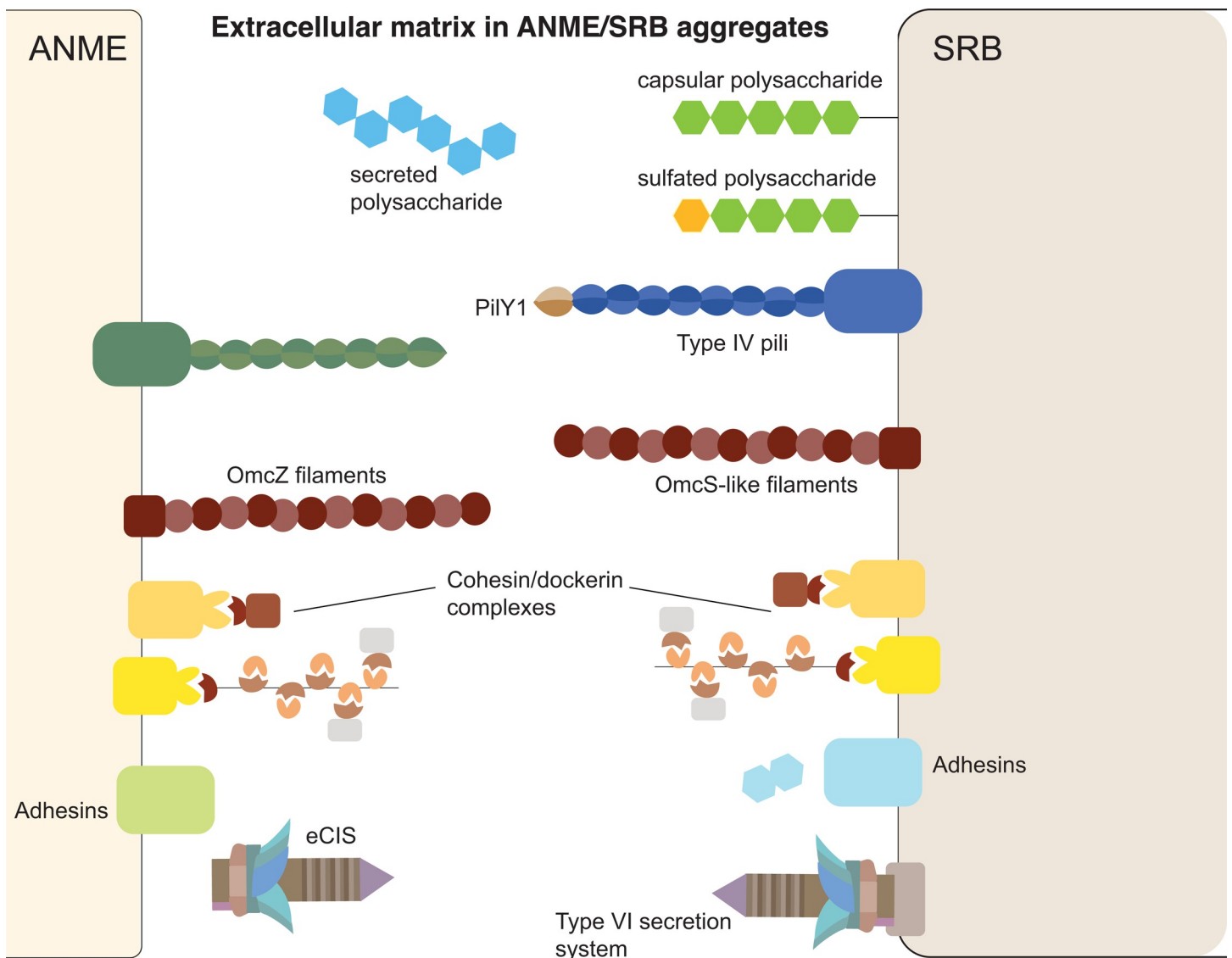

**Fig 6. Putative physiological factors involved in ANME/SRB aggregate formation.** Extracellular polysaccharides and protein complexes implicated in the formation of the extracellular matrix in ANME-SRB aggregates are visualized as cell-surface embedded or secreted. The capacity for biosynthesis of sulfated polysaccharides is present in 3 of the syntrophic SRB clades—Seep-SRB2, Seep-SRB1a, and Seep-SRB1g. Type VI secretion systems and eCISs are likely important for intercellular communication between ANME and SRB. ANME, anaerobic methanotrophic archaea; eCIS, extracellular contractile injection system; SRB, sulfate-reducing bacteria.

are conserved. This includes a neuraminic acid biosynthesis pathway, a sialic acid capsular polysaccharide widely associated with intestinal mucous glycans and used by pathogenic and commensal bacteria to evade the host immune system [112] (S14 Fig). These differences in polysaccharide biosynthesis pathways are likely reflected in the nature of the EPS matrix within each ANME-SRB aggregate.

Members of the thermophilic HotSeep-1 syntrophic SRB also encode for multiple putative polysaccharide biosynthesis pathways, including a pathway similar to the xap pathway in *G. sulfurreducens* (S15 Fig). The role of polysaccharides in the formation of conductive extracellular matrices and in intercellular communication is just beginning to be understood but they appear to be essential to its formation. For example, the mutation of the xap polysaccharide biosynthesis pathway in *G. sulfurreducens* eliminated the ability of this electrogenic bacteria to

reduce Fe (III) in the bacterium [102] and affected the localization of key multiheme cytochromes *c* OmcS and OmcZ and structure of the biofilm matrix [113], suggesting that the EPS matrix contributes a structural scaffold for the localization of the multiheme cytochromes. Similarly, the cationic polysaccharide pel in *P. aeruginosa* biofilms has recently been shown to play a role in binding extracellular DNA or other anionic substrates together forming tight electrostatic networks that provide strength to the extracellular matrix [114] and may offer a similar role in Seep SRB1a and 1g consortia. Based on the reported chemical composition of EPS from the Black Sea ANME-SRB biofilm [20], alongside TEM compatible staining of cytochromes *c* in the extracellular space between ANME and SRB [9,10,24], and the genomic evidence provided here of conserved polysaccharide biosynthesis pathways point to the existence of a conductive extracellular matrix within ANME-SRB consortia that has features similar to *Geobacter* biofilms [102]. While these conductive biofilms are correlated with the presence of secreted polysaccharides, the highly conserved capsular polysaccharides common in Seep SRB2 likely play a different role. In *Myxococcus xanthus*, the deletion of capsular polysaccharides leads to a disruption in the formation of multicellular fruiting bodies, suggesting a possible role for capsular polysaccharides in intercellular communication [115]. This is consistent with the universal role of O-antigen ligated lipopolysaccharides in cell recognition and the Seep SRB2 capsular polysaccharides may serve a similar purpose in consortia with ANME, either influencing within population interactions, or potentially mediating kin recognition.

**3.2 Several putative adhesins found in syntrophic SRB are absent in free-living SRB.** In addition to polysaccharides, there are several conserved adhesion-related proteins present in syntrophic SRB and absent in closely related SRB that are likely important for ANME-SRB biofilm formation. These include cohesin and dockerin domain-containing proteins, similar to those previously identified in ANME [14], immunoglobulin-like domains, cell-adhesin related domain (CARDB) domains, bacterial S8 protease domains, PEB3 adhesin domains, cadherin, integrin domains, and fibronectin domains (Fig 6 and S10 Table). Fibronectin domains are found in the one of the cytochromes *c*, oetF that is likely part of the EET conduit. This domain might interact with the conductive biofilm matrix itself or serve as a partnership recognition site. PilY1 is another adhesion-related protein that appears to be important in HotSeep-1. This is a subunit of type IV pili that is known to promote surface adhesion in *Pseudomonas* and intercellular communication in multispecies *Pseudomonas* biofilms [116]. Our analysis of the SRB adhesins suggests that some adhesins are conserved across a given syntrophic clade, while others appear to be more species or partnership specific. For example, while PilY1 is conserved across Seep-SRB2, the cohesin/dockerin complexes that are conserved in Hot-Seep1 and Seep-SRB1g are thus far found only in Seep-SRB2 sp. 4 and 8. Analysis of gene expression data suggest that in the Hot-Seep1/ANME-1 partnership, PilY1, an adhesin with an immunoglobulin-like domain and adjacent cohesin/dockerin domains might play a role in the syntrophic lifestyle [24] (S10 Table). In the ANME-2c/Seep-SRB2 partnership, PilY1, cohesin/dockerin complexes and a protein with a CARDB domain are highly expressed [24] (S10 Table). Curiously, in the Seep-SRB2 partnering with ANME-1, we could only identify 1 moderately expressed adhesin with a fibronectin domain [24] (S10 Table). We note the presence and high levels of expression of cohesin/dockerin domains in both ANME-2c and their verified Seep-SRB2 partner [24], and the presence of fibronectin domains in both ANME-2a and their Seep-SRB1a partner (S10 Table) suggesting that perhaps both partners within a partnership express and secrete similar kinds of extracellular proteins. This might serve as a mechanism for partnership sensing. While our analysis and that of earlier research into adhesins present in ANME [14] identify a number of conserved and expressed adhesins, further work is needed to investigate their potential role in aggregate formation.

**3.3 Secretion systems and intercellular communication in syntrophic SRB.** Extracellular contractile injection systems (eCISs) that resemble phage-like translocation systems (PLTSs) are found in some syntrophic SRB genomes (S11 Table and S16 Fig) although they are not as widely distributed as in ANME [14]. Typically, the eCIS bind to a target microorganism and release effector proteins into its cytoplasm. eCIS have been shown to induce death in worm larvae, induce maturation in marine tubeworm larvae [117] and found to mediate interactions between the amoeba symbiont and its host [118]. In ANME-SRB consortia, they might play a similar role with ANME releasing an effector protein into SRB, perhaps an effector molecule to promote the formation of a conductive biofilm or adhesins. Type VI secretion systems (T6SS) are similar to eCIS in facilitating intercellular communication between microorganisms. However, the primary distinction between them is that T6SS are membrane-bound while eCIS appear to be secreted to the extracellular space [119,120]. Interestingly, T6SS appear to be present in the ANME-2a partner Seep-SRB1a but absent in the ANME-2c partner Seep-SRB1a suggesting that they might play a role in mediating partnership specificity. While secretion systems are not uncommon in non-syntrophic bacteria, the high degree of their conservation in ANME and the high levels of expression of secretion systems in the ANME-2/Seep-SRB1a [25] and ANME-2c/Seep-SRB2 [24] partnerships suggest an important role for them in ANME-SRB syntrophy. Our analysis identified many conserved mechanisms for biofilm formation and intercellular communication in SRB to complement the pathways previously identified in ANME. Significantly, several polysaccharide biosynthesis pathways and adhesins were absent in the closest evolutionary neighbors of SRB indicating that adaptation to a syntrophic partnership with ANME required not just metabolic specialization but adaptation to a multicellular and syntrophic lifestyle.

## The adaptation of syntrophic SRB to partnerships with ANME

To better understand the evolutionary adaptations acquired by syntrophic SRB to form partnerships with ANME, we mapped the presence and absence of the above-mentioned pathways in central metabolism, nutrient sharing, biofilm formation, cell adhesion, and partner identification across each of the syntrophic SRB clades and their nearest evolutionary neighbors from the same bacterial order (S12 and S13 Tables). For example, the presence of the EET conduit OetABI in the Seep-SRB1a clade is nearly universal but, this trait is absent in the Desulfobacterales order that Seep-SRB1a belongs to, suggesting strongly that this machinery was horizontally acquired possibly in Seep-SRB1a or a closely related ancestor within the same family that includes Eth-SRB1. In contrast, most genomes in the order that Seep-SRB1g belongs to contain hydrogenases. However, hydrogenases are lacking in the syntrophic clade Seep-SRB1g implying that this trait was lost in the process of specialization to a partnership with ANME-2b. In addition to inferring adaptation based on presence and absence, phylogenetic trees were generated for at least 1 representative gene from each identified characteristic to corroborate the possibility of horizontal gene transfers (trees are available in S1 Data, Github (https://github.com/ranjani-m/syntrophic-SRB)). These trees provide further insight into the adaptation of various traits, the likely source of the genes received horizontally and in the case of Hot-Seep1 and Seep-SRB2 sp. 1 demonstrate the transfer of OetABI from one syntrophic clade to another. With the trees, we were able to also identify those genes that were vertically acquired but adapted for the respiratory pathways receiving DIET electrons, for example Tmc (S8 Fig). A brief summary of the gene gains and losses is provided in Fig 7 and S13 Table. Our analysis suggests that some traits are associated with partnerships with different ANME. The pel operon present in Seep-SRB1g and Seep-SRB1a is more closely associated with aggregates formed with the ANME-2a/b/c species rather than ANME-1. Similarly, the capsular

polysaccharide pseudaminic acid is present in those species of Seep-SRB1a that are associated with ANME-2c but absent in those species partnering ANME-2a suggesting that this polysaccharide might play a role in partnership identification and aggregate formation. Curiously, many of the adhesins we identified in the syntrophic SRB genomes have few close homologs in the NCBI NR dataset and almost no homologs in the nearest evolutionary neighbors (S13 Table), indicating that these proteins are likely highly divergent from their nearest ancestors. This is consistent with faster adaptive rates observed in extracellular proteins [121].

With our analysis, we identified many genes and traits that are correlated with a syntrophic partnership with ANME, but it is less easy to identify whether they are essential. The complete conservation of the OetI-type or other EET cluster (such as OmcKL) suggests these are essential, but not sufficient, for the formation of this partnership since the multiheme cytochrome conduits themselves are present in many organisms not forming a syntrophic partnership with ANME. There is also a strong signature for the presence of a secreted polysaccharide pathway such as the pel operon in Seep-SRB1a and Seep-SRB1g and a xap-like polysaccharide in Hot-Seep1 and Seep-SRB2. With these components, a conductive biofilm matrix can be established, but the means of partnership recognition and communication between the archaea and bacteria are less clear. As suggested previously [14], the near complete conservation of the eCISs in ANME might play a role in partnership identification. The target receptor of the eCIS is unclear but the presence of conserved capsular polysaccharides in SRB that often are the target of bacteriophages and pathogens is suggestive as a possible site for binding. Likewise, the high levels of expression of cohesin and dockerin complexes by both ANME and SRB in the ANME-2c/Seep-SRB2 partnership are indicative of a role in syntrophic partnership [24]. In Seep-SRB1a, there are conserved fibronectin domains that likely bind the biofilm matrix and Seep-SRB2 has a conserved cell-surface protein with a PEGA sequence motif (S12 Table).

We can infer something about the order of evolutionary adaption of syntrophic SRB from what is essential and conserved in syntrophic SRB and what is present in their nearest evolutionary ancestors. The presence of DIET complexes such as OetABI in the nearest evolutionary neighbors of HotSeep-1 (Desulfofervidales), Seep-SRB2 (Dissulfuribacteriales), and Seep-SRB1g (Seep-SRB1c) and the absence of adhesins (cohesins) and polysaccharide biosynthesis (pel) in the related clades (Fig 7) suggests that the acquisition of DIET pathways in an ancestral clade was the first and essential step towards adaptation towards a syntrophic lifestyle. Then, the syntrophic partners likely acquired the pathways needed for aggregate formation (such as adhesins, the pel polysaccharide biosynthesis pathway) after. Seep-SRB2 contains a respiratory trait (CbcBA) that is absent in its nearest evolutionary neighbor (Fig 4). This indicates that more steps were required for the adaptation of this clade to a syntrophic partnership with ANME. The greater diversity within the clades Seep-SRB1a and Seep-SRB2 may be a result of the larger number of partnerships with different ANME compared to a clade such as Seep-SRB1g. However, there is insufficient evidence to rule out the possibility of promiscuous partnership formation with multiple ANME within each SRB species. In these cases, the observed species diversity within Seep-SRB1a and Seep-SRB2 must be driven by other factors. Our analysis shows that the adaptation towards EET and the formation of conductive biofilms was likely driven by a greater selection pressure than the adaptation to a specific ANME partner. Consistent with this, the gain and loss of specific adhesin and matrix-binding proteins is more dynamic.

Another aspect of the adaptation of syntrophic SRB is the high number of inter-clade transfers. In addition to the likely transfer of OetABI between HotSeep-1 and Seep-SRB2 sp.1 (S7 Fig), we also note a high degree of similarity between the proteins of the following components in different clades of syntrophic SRB—cohesin/dockerin modules, the OmcKL conduit, and enzymes in the pel and xap polysaccharide biosynthesis pathways. These appear to be the result

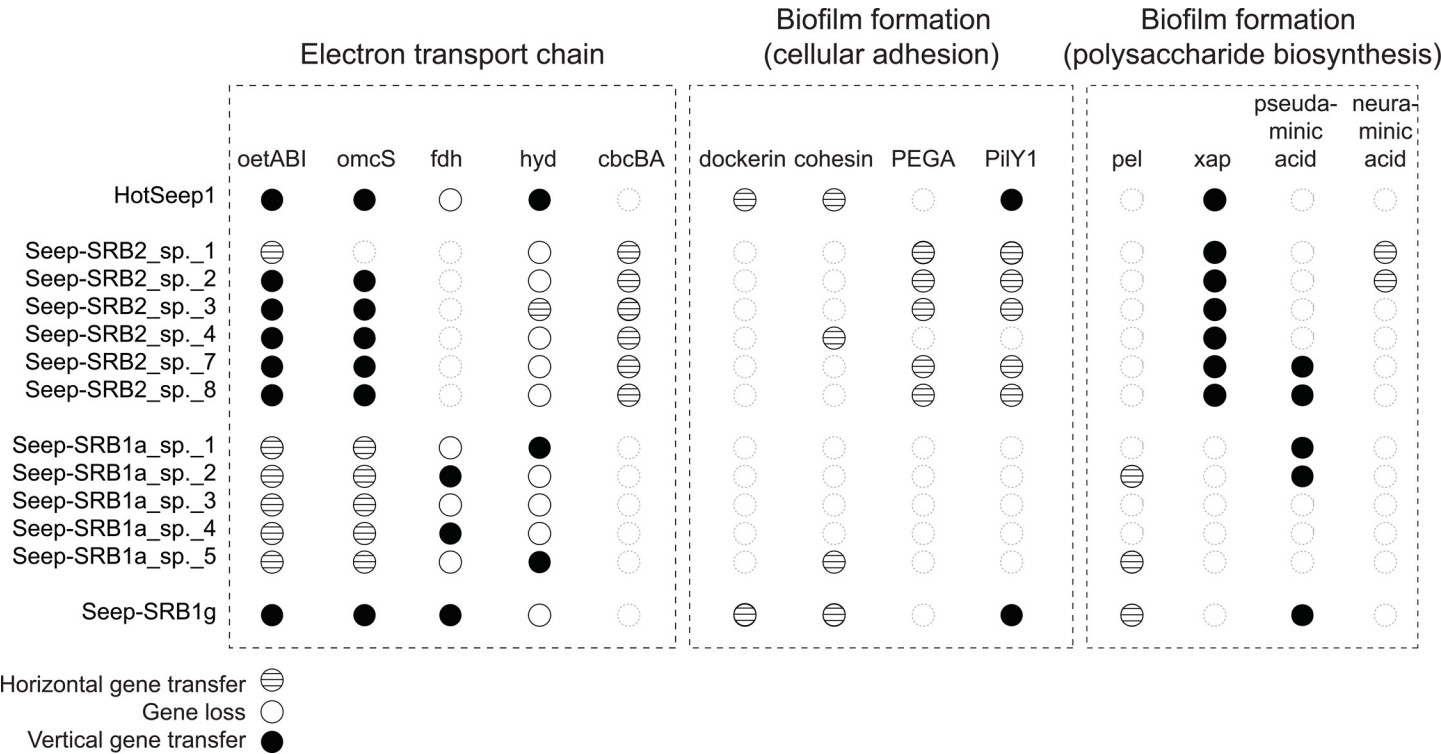

**Fig 7. A summary of important gene loss and gain events in the physiological adaptation of sulfate reducing bacteria that led to a syntrophic partnership with ANME.** The presence and absence of genes involved in the electron transport chain, nutrient sharing, biofilm formation, and cellular adhesion are listed in S12 Table. We identified genes that were potentially gained, lost, or biochemically adapted using a comparative analysis of the presence a given gene in a syntrophic clade in its order-level taxonomic background. For example, if a gene is present in a syntrophic SRB clade and is present in fewer than 30% of the remaining species in a given order, this gene is considered a likely horizontally transferred gene. The likelihood of horizontal transfer is then further corroborated with a phylogenetic tree of that gene generated with close homologs from NCBI and our curated dataset. The trees are available in S1 Data. The secondary analysis of the likelihood of gene gains and losses is present in S13 Table. ANME, anaerobic methanotrophic archaea; SRB, sulfate-reducing bacteria.

of inter-clade transfers and the high number of transfers might imply that a mechanism promoting the exchange of DNA exists in this environment between ANME and SRB, either through a viral conduit or perhaps with the eCIS carrying DNA as cargo. Further analysis is needed to identify the number of transfer and the sources of transfers. In fact, a thorough accounting of these horizontal gene transfers combined with molecular clock dating might provide insight into the timeline and the relative age of the different ANME/SRB partnerships. Our phylogenomic analysis places the verified ANME-2c partners as ancestral to the ANME-2a partners within the Seep-SRB1a clade (Figs 1 and S1). Within the Seep-SRB2 clade, the topology places an ANME-1 partner as basal to the remaining Seep-SRB2 and the only verified ANME-2c partner as one of the later branching members (Figs 1 and S1). Earlier research places ANME-1 as the deepest branching lineage of ANME [14] and this relative ancestry of partners might suggest that Seep-SRB2 is older than Seep-SRB1a. However, it appears that ANME-1 acquired its mcr through horizontal gene transfer [14], and we have insufficient data to know when this occurred. Thus, we cannot know that ANME-1 was methanotrophic when it diverged from the Methanomicrobiales. These observations suggest that we cannot constrain the emergence of AOM solely through the relative branching patterns of the various ANME and SRB clades. A more thorough reconstruction of the adaptive gene transfers using the framework established for ANME and in this work for syntrophic SRB would provide insight into the evolution of this biogeochemically important syntrophic partnership.

## Conclusions

This comparative genomic analysis of the major ANME-partnering SRB clades provides a valuable metabolic and evolutionary framework to understand the differences between the various syntrophic sulfate reducing partners of anaerobic methanotrophic archaea and develop insight into their metabolic adaptation. In this work, we show that the electron transport chains of the different syntrophic SRB partners of ANME are adapted to incorporate EET conduits that are needed for DIET. Groups including the Seep-SRB2 appear to have acquired cytoplasmic membrane complexes that can function with the EET conduits, while Seep-SRB1a clades have adapted existing inner-membrane complexes for interaction with the EET conduit. Electron bifurcation also appears to be common across the syntrophic lineages and is often coupled to the cytoplasmic machinery and likely provides an advantage in low-energy environments. We also show that the coevolution between different ANME and SRB partners may have resulted in nutritional interdependencies, with cobalamin auxotrophy observed in at least one of the specific syntrophic SRB subclades. Our genome-based observations provide insight into the various adaptations that are correlated with the formation of different ANME-SRB partnerships. These adaptive traits appear to be related with mechanisms driving other ecological phenomena such as biofilm formation and non-obligate syntrophic interactions. The identification of these traits allowed us to posit important steps in the evolutionary trajectory of these SRB to a syntrophic lifestyle. While the full import of these observations is not yet clear, they offer a roadmap for targeted physiological investigations and phylogenetic studies in the future.

## Materials and methods

### Ethics statement

Our environmental samples constituted sediment samples collected from methane seeps and hydrothermal vents. We followed protocol for ethical sampling and received appropriate permission for collection of environmental samples when required. Sample collection and export permits for methane seep sediment samples off the coast of Costa Rica were acquired through the Costa Rican Ministry of the Environment and Energy (SINAC-CUSBSE-PI-R-032-2018 and Academic License SINAC-SE-064-2018). Sample collection permits for FK181031 (25/07/2018) were granted by la Dirección General de Ordenamiento Pesquero y Acuícola, Comisión Nacional de Acuacultura y Pesca (CONAPESCA: Permiso de Pesca de Fomento No. PPFE/DGOPA-200/18) and la Dirección General de Geografía y Medio Ambiente, Instituto Nacional de Estadística y Geografía (INEGI: Autorización EG0122018), with the associated Diplomatic Note number 18–2083 (CTC/07345/18) from la Secretaría de Relaciones Exteriores - Agencia Mexicana de Cooperación Internacional para el Desarrollo/Dirección General de Cooperación Técnica y Científica. Sample collection permit for cruise NA091 (18/04/2017) was obtained by the Ocean Exploration Trust under permit number EG0072017.

### Sampling locations and processing of samples

Push-core samples of seafloor sediment were collected from different locations on the Costa Rica margin during the AT37-13 cruise in May 2017 (sample serial numbers: #10073, #9063), southern Pescadero Basin [122] during the FK181031 cruise on R/V Falkor operated by the Schmidt Ocean Institute in November 2018 (sample serial number: #PB10259, live incubation of the top 3 cm section of push core #FK181031-S193-PC3, 123], and from Santa Monica Basin during WF05-13 cruise in May 2013 (sample serial numbers: #7059). Sediment push-cores retrieved from the seafloor were sectioned into 1 to 3 cm sediment horizons. At the time

of shipboard processing, approximately 2 mL of the sediment was sampled for DNA extraction and FISH analysis and the rest was saved in Mylar bags under an $N_2$ atmosphere at 4˚C for sediment microcosm incubations. Microbial mat sample #14434 was collected from Santa Monica Basin during the WF02-20 cruises in February 2020. Rock samples were retrieved from South Pescadero Basin [123] during the NA091 cruise on E/V Nautilus operated by the Ocean Exploration Trust in October to November 2017 and the FK181031 cruises (sample serial numbers: NA091-R045, NA091-R008, and #12019, #11946, #11719, respectively) and saved in Mylar bags under an $N_2$ atmosphere at 4˚C.

Sediment horizons from samples 10073, 9063, and 7059 were incubated in artificial sea water as previously described [25,36] with $CH_4$ and 250 μM L-Homopropargylglycine (HPG) at 4˚C. Once the presence of metabolically active ANME-SRB in these microcosms was confirmed by the accumulation of sulfide combined with observation of incorporation of HPG by BioOrthogonal Non-Canonical Amino Acid Tagging (BONCAT), samples from these incubations were used for sorting of single-aggregates by FACS as described below. #FK181031-S193-PC3 was incubated in anaerobic artificial sea water without electron donor at 24˚C. Rock #NA091-R045 was incubated in anaerobic artificial sea water supplemented with pyruvate at 24˚C. Rock samples from S. Pescadero Basin (#11946, #11719, and #12019) were also incubated with artificial sea water and $CH_4$ at 50˚C.

## DNA extraction followed by metagenome sequencing for samples #11946, #11719, #12019, #NA091-R045, #NA091-R008, #PB10259, and #14434

For incubations of carbonate samples #11946, #11719, and #12019, DNA was extracted from approximately 500 mg of crushed rock samples using a modified version of the Zhou protocol [124] as follows. Prior to the incubation with proteinase K, the sample was incubated with lysozyme (10 mg ml-1) for 30 min at 37˚C; 10% SDS was used for incubation; after SDS incubations, the sample was extracted twice by adding 1 volume (1 mL) of phenol/chloroform/isoamylalcohol (25:24:1) with incubation for 20 min at 65˚C followed by centrifugation; in the final step, the DNA was eluted in 40 μl of TE 1× buffer. Approximately 250 mg of sediment sample #PB10259 and microbial mat sample #14434 were extracted using the QIAGEN Power Soil Kit, and 500 mg of crushed carbonate samples #NA091-R045, #NA091-R008 were also extracted using the QIAGEN Power Soil Kit.

For samples #PB10259, #14434, NA091-R045, DNA libraries were prepared using the NEBNext Ultra kit and sequenced at Novogene with the instrument HiSeq4000. A library was also prepared using the NEBNext Ultra kit for NA091-008. This sample was sequenced at Quick Biology (Pasadena, California, United States of America) with a HiSeq2000 using a 2 × 150 protocol. DNA libraries for samples #11946, #11719, and #12019 were prepared using the Nextera Flex kit and also sequenced at Novogene on the HiSeq4000. After sequencing of NA091-R45, primers and adapters were removed from all libraries using bbduk [125] with mink = 6 and hdist = 1 as trimming parameters and establishing a minimum quality value of 20 and a minimal length of 50 bp.

## Assembly and binning of metagenomes from samples samples #NA091-R045, #NA091-R008, #PB10259, #14434, #11946, #11719, and #12019

Metagenomes from samples #PB10259, #14434, #11946, #11719, and #12019 were assembled individually using SPAdes [126] v3.14.1, and each resulting assembly was binned using metabat v2.15 [127]. Automatic prediction of function for genes within the various MAGs was performed using prokka v.1.14.6 [128]. The reads of the DNA libraries derived from the rock

sample (NA091-R008) were assembled individually using SPAdes v.3.12.0. From the de novo assemblies for NA091-R008, we performed manual binning using Anvio v.6 [45]. We assessed the quality and taxonomy affiliation from the obtained bins using CheckM [129] and GTDB-tk [130]. Genomes of interest affiliated to Desulfobacterota were further refined via a targeted-reassembly pipeline. The trimmed reads for the NA091-008 assembly were mapped to the bin of interest using bbmap [125] (minimal identity of 0.97), then the mapped reads were assembled using SPAdes and finally the resulting assembly was filtered discarding contigs below 1,500 bp. This procedure was repeated for 13 to 20 cycles for each bin, until the bin quality did not improve any further. Bin quality was assessed based on the completeness, contamination (<5%), N50 value, and number of scaffolds of the bin using CheckM. The resulting bins were considered as MAGs. Automatic prediction of function for genes within the various MAGs was performed using prokka v.1.14.6 [128] and curated with the identification of Pfam [131] and TIGRFAM [132] profiles using HMMER v.3.3 [133]; KEGG domains [134] with Kofam [135] and of COGs and arCOGs motifs [136] using COGsoft [137].

## Fluorescent-sorting of metabolically active single aggregates from samples #10073, #9063, and #7059 followed by sequencing

Sediment-extracted consortia from samples #10073, #9063, and #7059 were analyzed. Individual ANME:SRB consortia were identified and sorted using fluorescent signal, as previously described [36]. The SYBR-Green dye was excited using a 488-nm laser, and fluorescence was captured with a 531-nm/30-nm filter. Gates were defined using a forward scatter (FSC) versus 531-nm emission plot, and events with a fluorescent signal brighter than >90% of aggregates in the negative control were captured. For sample #10073, 50 consortia were sorted into 1.5 mL tubes and stored at 4˚C for sequencing. For samples #9063 and #7059, 28 and 19 consortia were sorted, respectively.

Single consortia were lysed and DNA was amplified using multiple displacement amplification (MDA) protocol as previously described [138]. The amplified DNA was sheared, attached to Illumina adapters, and sequenced using the Illumina NextSeq-HO method. Only metagenomes from 2, 3 and 1 sorted aggregates from each of the samples# 10073, #9063, and #7059 respectively were used in this study.

## Assembly and binning of single aggregate metagenomes from samples #10073, #9063, and #7059

Metagenomes were assembled using SPAdes spades v. 3.13.0 and annotated using the integrated microbial genomes (IMG) annotation pipeline. As the single aggregate metagenomes represent extremely reduced communities, and the MDA precludes traditional contig binning by coverage, the metagenomes were binned using a manual approach based on sequence composition and taxonomic assignment of the genes. Manual binning was performed using a principal component analysis (PCA) of the tetramer frequency of the contigs, calculated using calc.kmerfreq.pl (https://github.com/MadsAlbertsen/miscperlscripts/). Taxonomic affiliation of the genes on the contigs was taken from the IMG annotation, and the percentage of genes on each contig annotated as "Archaeal" or "Bacteria" was used to corroborate the clustering in the PCA plot. Jupyter notebooks used for the binning are available at https://github.com/dspeth/SRB_single_aggregate_bins. Automatic metabolic prediction of the MAGs was performed using prokka v.1.14.6 [128].

## Taxonomic classification of metagenome bins from various syntrophic sulfate reducing bacteria

Single copy marker genes identified in the "Bacteria 71" gene set included in Anvio [120] were extracted from each of the syntrophic SRB genomes and all genomes within the phylum Desulfobacterota available in release 95 of the Genome Taxonomy Database [38]. A concatenated gene alignment was generated using MUSCLE [72] as part of the anvio script "anvi-get-sequences-for-hmm-hits." A phylogenetic tree was inferred using FastTree as per the Anvio-7 pipeline using the command "anvi-gen-phylogenomic-tree," in order to provide a phylogenetic context for each of the 4 SRB clades. We corroborated our phylogenetic placement with the classification provided by GTDB-tk [130]. Additionally, we assessed the extent of taxonomic diversity within the 4 clades by calculating the ANI and 16S rRNA sequence similarity between different organisms that belong to each clade using PyANI [139] in Anvio-7 [45]. A 95% ANI value of 95% [140] and 98.65% similarity in 16S rRNA [141] were used as cut-offs to delineate different species.

## Phylogenetic analysis of OetI, the outer-membrane beta barrel forming protein in the OetI-type cluster

OetI here refers specifically to the outer-membrane beta barrel forming protein in the OetI-type cluster implicated in DIET between ANME and SRB. All OetI sequences were identified in the genomes of the syntrophic SRB clades by using BLASTP [142] with the query OEU57520.1 from Seep-SRB1g sp. C00003106 and an e-value of e-30. When no OetI hits were found in the syntrophic SRB genomes using this query, we tested for the existence of a beta barrel within 10 genes of every multiheme cytochrome that contained more than 5 heme *c* binding motifs using PRED-TMBB [143]. In this way, we identified 17 EET gene clusters in the syntrophic SRB genomes and 5 clusters from non-syntrophic Seep-SRB1c, Desulfofervidales and Dissufuribacterales. Protein sequences of OetI from each of these clusters were used as queries to extract all the closest homologs for each of these OetI sequences from the NCBI database. This search was performed using BLASTP with an e-value cut of 1e-5. The extracted sequences were aligned and manually curated to eliminate sequences that were too short and to remove nonspecific hits. A phylogenetic tree was inferred using IQ-TREE2 [73], a Dayhoff model of substitution and 1,000 ultrafast bootstrap iterations and visualized using the iTOL web server [144].

## Phylogenetic analysis of other respiratory proteins

All sequence alignments used for analysis of respiratory proteins were made using MUSCLE [72] and visualized using Jalview [145]. Phylogenetic trees of all proteins were inferred using IQ-TREE2 [73] except for the following - OetB, omcX, TmcD, and TmcA. Phylogenetic trees for OetB, omcX, TmcD, and TmcA were inferred using RAxML [146]. RAxML trees were inferred using a Dayhoff model of rate substitution and 100 bootstraps. IQ-TREE trees were inferred using 1,000 ultrafast bootstraps while the models were automatically selected by IQ-TREE using the Bayesian information criterion (BIC). The models used for each specific tree are available in S14 Table.

## Sequences analysis of all cytochromes *c*

All cytochromes *c* were identified from the MAGs of syntrophic SRB by employing a word-search method with a custom python script by querying for the commonly found "CxxCH" motif in cytochromes *c*. Once these sequences were extracted, they were aligned using

MUSCLE [72]. Clusters were identified depending on the presence of well-defined regions using visual inspection. The clusters were then tabulated in **S6 Table**. The cellular localization of cytochromes *c* was inferred either from the cellular localization of homologous cytochromes *c* from Desulfuromonadales or using Signal P-5.0 [147].

### Sequences analysis of all putative adhesins

Adhesins were identified from the MAGs of syntrophic SRB by using the "all-domain" annotation feature on KBase as previously described [14,148]. Once putative domains were predicted, we extracted the coding features that corresponded to all putative adhesins based on searches for the words "integrin," "adhesin," "cohesin," "dockerin," "fibronectin," "PilY," and "immunoglobulin in the domain descriptions." The proteins corresponding to these results were aligned using MUSCLE [72]. Adhesin clusters were identified depending on the presence of well-defined regions using visual inspection and then additionally verified by use of the NCBI Conserved Domain database [149]. The adhesins were then tabulated in S10 Table.

### Identification of putative polysaccharide biosynthesis pathways

Once, the syntrophic SRB genomes were annotated using the Prokaryotic Genome Annotation Pipeline (PGAP) [150], we identified polysaccharide biosynthesis pathways by looking for the presence of glycosyl transferases, aminotransferases, sugar transporters, and polysaccharide biosynthesis proteins. If gene cassette structures followed known operon structures of ABC transporter-type, Wzx/Wzy, or synthase type pathways [151], they were retained and tabulated in S9 Table and visualized in S12–S15 Figs.

### Evolutionary analysis of important genes to identify gains, loss, and biochemical adaptation

We tabulated the presence and absence of 33 traits that we propose are important for the formation of ANME-SRB partnership in each syntrophic SRB clade and the taxonomic order from which they originate. The presence and absence was identified using BLASTP searches with a query sequence or HMM as listed in S13 Table. If a gene is present in over 30% of non-syntrophic relatives in a given order, it is considered as present in this order or in a syntrophic SRB clade, it is considered present in this taxonomic clade. If a gene is present in the order and in the syntrophic SRB clade that belongs to this order, the gene is considered to be vertically transferred. If a gene is present in the order but, absent in the syntrophic SRB, the gene is considered to be lost. If a gene is present in the syntrophic SRB but absent in the order it belongs to, the gene is considered to be horizontally acquired. The last assumption is corroborated as much as possible by gene trees deposited on Github (https://github.com/ranjani-m/syntrophic-SRB).

### Supporting information

**S1 Fig. Taxonomic separation of syntrophic sulfate reducing bacteria using average nucleotide identity.** The average nucleotide identity of genomes from each clade of the syntrophic sulfate reducing bacteria and some related bacteria were computed using the PyANI program available through Anvi'o [45]. The different clades of syntrophic SRB—HotSeep1, Seep-SRB2, Seep-SRB1a, and Seep-SRB1g are colored according to the attached legend. The Seep-SRB1a genomes in particular are differently colored depending on whether they partner ANME-2a or ANME-2c, respectively. The geographic location from which each genome was extracted is

indicated on each node or clade in the tree.
(EPS)

**S2 Fig. Comparison of 16S rRNA and 23S rRNA phylogeny of organisms from the phylum Desulfobacterota.** 16S rRNA and 23S rRNA sequences were extracted from all organisms from the phylum Desulfobacterota available in GTDB release 95 [38] and from syntrophic SRB. These sequences were aligned using MUSCLE [72] and a tree was inferred using IQTREE2 [73]. Hot-Seep1 is placed adjacent to the order Thermodesulfbacteriales in both these trees. The trees are available in Newick format in S2 Data.
(EPS)

**S3 Fig. Phylogeny of RpoB from organisms within the phylum Desulfobacterota.** Sequences of RNA Polymerase, subunit B were extracted from all organisms from the phylum Desulfobacterota available in GTDB release 95 [39] and from syntrophic SRB using BLASTP [142] with an e-value cut-off of e-30 and appropriate query sequences. The sequences were confirmed to RpoB by manual inspection of a multiple sequence alignment generated using MUSCLE [72] and the tree was inferred using IQTREE2 [73]. In this tree, Hot-Seep1 is found adjacent to the order Desulfovibrionales. The tree is available in Newick format in S2 Data.
(EPS)

**S4 Fig. Placement of various Seep-SRB1 clades within the phylum Desulfobacterota.** A phylogenetic tree of full-length 16S sequences from various Seep-SRB1 clades including the original 16S rRNA sequences [23] used to define the Seep-SRB1(a–f) clades.
(EPS)

**S5 Fig. Pan-genome analysis of Seep-SRB1a metagenomes.** Fourteen genomes from nine Seep-SRB1a species were analyzed using the Anvi'o pan-genome analysis pipeline [45]. Five gene cluster bins were annotated based on genes that were identified as part of the core meta-genome, unique to Seep-SRB1a sp. 2, Seep-SRB1a sp. 3, Seep-SRB1a sp. 5, and from the Seep-SRB1a sp. 4 and 7.
(PNG)

**S6 Fig. Pan-genome analysis of Seep-SRB2 metagenomes.** Nineteen genomes from eight Seep-SRB1a species were analyzed using the Anvi'o pan-genome analysis pipeline [45]. Three gene cluster bins were annotated based on genes that were identified as part of the core meta-genome, present in Seep-SRB2 sp. 1 and absent in Seep-SRB2 sp. 1.
(PNG)

**S7 Fig. Phylogenetic placement of the outer membrane beta barrel, OetI from the putative DIET cluster.** A multiple sequence alignment, Supplementary multiple sequence alignment MSA2 of the OetI protein sequences extracted from the genomes of syntrophic SRB and the NCBI database was generated using MUSCLE [72]. This alignment was used to infer a phylogenetic tree using IQ-Tree2 [73] and visualized on the iTOL web server [144]. (a) The phylogenetic placement of OetI from E20 Seep-SRB2 next to OetI from Thermodesulfobacteria and *Dissulfurirhabdus thermomarina* demonstrates that it was possibly vertically acquired from a gene transfer that was ancestral to the Seep-SRB2 and then vertically transferred. (b) The phylogenetic placement of Seep-SRB1a and Seep-SRB1g OetI suggests that they are related. Additionally, the placement of OetI from G37 Seep-SRB2 next to OetI from Desulfofervidales suggests that the Seep-SRB2 partner of ANME-1 acquired its DIET cluster from HotSeep-1. The phylogenetic tree of OetI is available in Newick format, and the presence/absence table of OetI in Desulfobacterota is also available in S2 Data.
(EPS)

**S8 Fig. The Tmc complex in Seep-SRB1a and Seep-SRB1g are divergent.** (A) The operons containing Tmc in *Candidatus Desulfofervidus auxilii* (HotSeep-1), Seep-SRB1g, and Seep-SRB1a show that TmcA is present in the former operon while it is missing in the latter 2. (B) The distribution of Tmc is mapped on to the phylum Desulfobacterota, showing that it is common in the order Desulfobacterales and Desulfovibrionales. (C) Phylogeny of TmcD demonstrates that the Tmc complex in Seep-SRB1a and Seep-SRB1g cluster together and appear to be different from the Tmc complex in other organisms from the orders Desulfobacterales and C00003060. The phylogenetic tree of TmcD is available in Newick format, and the presence/absence table of Tmc in Desulfobacterota is also available in S2 Data.
(EPS)

**S9 Fig. Distribution of QrcABCD and RnfABCDEG in Desulfobacterota.** The presence and absence of Qrc and Rnf was demonstrated across the Desulfobacterota using BLASTP [142] searches of different query sequences of these complexes. Both these complexes are absent from the orders Desulfobulbales, Thermodesulfobacteriales, and Dissulfuribacteriales. The presence/absence table of Qrc/Rnf in Desulfobacterota is also available in S2 Data.
(EPS)

**S10 Fig. Gene neighborhoods of various HdrA containing complexes and carbon fixation pathways in syntrophic SRB.** HdrA homologs were identified in the 4 syntrophic SRB clades with the following gene neighborhoods. Multiple HdrA homologs were identified adjacent to the carbon fixation pathways in syntrophic SRB, specifically Seep-SRB1a and Seep-SRB1g.
(EPS)

**S11 Fig. Operons of Flx-Hdr complexes and gene neighborhood of putative formate utilizing proteins.** (a) Flx-Hdr complexes that recycle electrons between NADH, ferredoxins, and DsrC are found in Seep-SRB1g, Seep-SRB1a, and Seep-SRB2. (b) Many putative formate utilizing proteins are found in Seep-SRB1g, Seep-SRB1a, and Seep-SRB2. In Seep-SRB1g and Seep-SRB1a, periplasmic formate dehydrogenases (fdhAB) are found. fdhA domains as identified here are typically found in the periplasm and have a respiratory function while fdhF2 are typically cytoplasmic.
(EPS)

**S12 Fig. Putative polysaccharide biosynthesis pathways in Seep-SRB1a.**
(PNG)

**S13 Fig. Putative polysaccharide biosynthesis pathways in Seep-SRB1g.**
(EPS)

**S14 Fig. Putative Polysaccharide biosynthesis pathways in Seep-SRB2.**
(PNG)

**S15 Fig. Putative polysaccharide biosynthesis pathways in HotSeep-1.**
(PNG)

**S16 Fig. Presence of extracellular contractile injection systems (eCIS) in ANME and SRB.** The presence of eCIS conduits in ANME and SRB was identified using BLASTP [142] and dbeCIS [119]. While the eCIS clusters are widely distributed in ANME-2a and ANME-2b, they are only sparsely distributed in ANME-1. They are only present in the Seep-SRB1g species found in Costa Rica and one of the Seep-SRB1a species from Santa Monica Basin. Lines in green and light teal are used to depict the partnerships between ANME-2c and verified species of Seep-SRB1a, and ANME-2a and verified species of Seep-SRB1a, respectively. Line in blue is

used to depict the partnership between ANME-2b and Seep-SRB1g. Underlying presence/absence data is available in S2 Data.
(EPS)

**S17 Fig. Phylogeny of afp10, the spike protein from the extracellular contractile injection system.** Afp10 is the PAAR-domain containing protein that typically interacts with the target organism of the eCIS. Afp10 sequences were extracted from ANME and SRB using BLASTP [142] and dbeCIS [119]. These sequences were then used as queries to repeatedly search and identify the closest homologs from the NCBI database. A sequence alignment was then made using MUSCLE, manually inspected and filtered, and the tree was inferred using RAxML [146]. Seep-SRB1a sequences are related to other eCIS sequences from Desulfobacterales while Seep-SRB2 afp10 and Seep-SRB1g afp10 sequences do not cluster with evolutionarily related bacteria. The phylogenetic trees of afp10 are available in Newick format in S2 Data.
(EPS)

**S18 Fig. Phylogeny of afp11, a base plate protein from the extracellular contractile injection system.** Afp11 belongs to the baseplate of the eCIS and does not interact directly with the target organism. Afp11 sequences were extracted from ANME and SRB using BLASTP [142] and dbeCIS [119]. These sequences were then used as queries to repeatedly search and identify the closest homologs from the NCBI database. A sequence alignment was then made using MUSCLE, manually inspected and filtered, and the tree was inferred using RAxML [146]. Seep-SRB1a sequences are related to other eCIS sequences from Desulfobacterales while Seep-SRB2 afp10 and Seep-SRB1g afp10 sequences do not cluster with evolutionarily related bacteria. The phylogenetic trees of afp11 are available in Newick format in S2 Data.
(EPS)

**S19 Fig. Structural model of TmcD from Seep-SRB1a.** A structural model of TmcD from Seep-SRB1a was generated using Alphafold2 [152] using the monomer option. This model was superimposed on top of a structural model of TmcD from Olavius algarvensis available on UniProt. The divergent sequence regions from TmcD were highlighted in red while cysteine residues unique to Seep-SRB1a were highlighted in green. The conserved residues identified here were observed using a multiple sequence alignment of TmcD made available in online supplementary data.
(PNG)

**S1 Table. List of genomes from the Genome Taxonomy Database [38] used for comparative analysis in this study.**
(XLSX)

**S2 Table. 16S rRNA pairwise similarity matrix.** A pairwise comparison matrix of 16S rRNA sequence similarity between syntrophic SRB clades and their nearest evolutionary relatives was generated using Anvi'o [45].
(XLSX)

**S3 Table. Seep-SRB1a pan-genome analysis.** A pan-genome analysis of Seep-SRB1a clades was performed using Anvi'o [45] to demonstrate the conserved genes and differences within the syntrophic SRB clade Seep-SRB1a.
(XLSX)

**S4 Table. Seep-SRB2 pan-genome analysis.** A pan-genome analysis of Seep-SRB1a clades was performed using Anvi'o [45] to demonstrate the conserved genes and differences within the

syntrophic SRB clade Seep-SRB2.
(XLSX)

**S5 Table. List of accession numbers for operons containing the OetI-type conduit in syntrophic SRB and their nearest evolutionary neighbors.**
(XLSX)

**S6 Table. List of protein accession numbers for cytochromes *c* identified in syntrophic SRB and their nearest evolutionary neighbors.** Cytochromes *c* clusters were identified by clustering with multiple sequence alignment [72] and classified as playing roles in extracellular electron transfer, periplasmic electron transfer, or inner membrane electron transfer.
(XLSX)

**S7 Table. List of protein accession numbers for respiratory genes and nitrogen fixation in syntrophic SRB and other organisms within Desulfobacterota.**
(XLSX)

**S8 Table. List of protein accession numbers for cobalamin biosynthesis pathways in syntrophic SRB and their nearest evolutionary neighbors.**
(XLSX)

**S9 Table. List of protein accession numbers for polysaccharide biosynthesis pathways in syntrophic SRB and their nearest evolutionary neighbors.**
(XLSX)

**S10 Table. List of protein accession numbers for putative cellular adhesins in syntrophic SRB and their nearest evolutionary neighbors.**
(XLSX)

**S11 Table. List of protein accession numbers of extracellular contractile injection systems in syntrophic SRB and ANME.**
(XLSX)

**S12 Table. List of selected genes from different pathways involved in extracellular electron transfer, respiratory pathways, and biofilm formation present and absent in syntrophic SRB and their nearest evolutionary neighbors.**
(XLSX)

**S13 Table. List of genes expected to be gained by horizontal gene transfer, vertical inheritance or lost, based on patterns of presence and absence in syntrophic SRB and their nearest evolutionary neighbors. Also supported by manual inspection of trees made available on Github (https://github.com/ranjani-m/syntrophic-SRB).**
(XLSX)

**S14 Table. List of protein phylogenetic models used for inference of evolutionary patterns within proteins in this study.**
(XLSX)

**S1 Text.** (S1_text_Proposal_for_formal_nomenclature.pdf).
(PDF)

**S1 Data.** Gene_trees_from_syntrophic_SRB.zip.
(ZIP)

**S2 Data.** Other supplementary data.zip.
(ZIP)

## Acknowledgments

We would like to acknowledge Magdalena Mayr for her thoughtful comments on this manuscript and Fernanda Jimenez-Otero for sharing her expertise in the field of extracellular electron transfer. We are also grateful to Alon Philosof, Aditi Narayan, Kriti Sharma, and James Hemp for many productive discussions that assisted in the framing of this manuscript. We are indebted to the pilots, crew, and shipboard scientists on the *R/V Western Flyer* (Monterey Bay Research Aquarium Institute), *R/V Falkor* (Schmidt Ocean Institute), *E/V Nautilis* (Ocean Exploration Trust), and *R/V Atlantis* whose dedication and skills made this research possible.

## Author Contributions

**Conceptualization:** Ranjani Murali, Hang Yu, Kyle S. Metcalfe, Grayson L. Chadwick, Victoria J. Orphan.

**Data curation:** Ranjani Murali, Daan R. Speth, Fabai Wu, Rex R. Malmstrom, Danielle Goudeau, Tanja Woyke, Roland Hatzenpichler, Stephanie A. Connon.

**Formal analysis:** Ranjani Murali, Daan R. Speth, Kyle S. Metcalfe, Rafael Laso-Pèrez, Rex R. Malmstrom.

**Funding acquisition:** Rex R. Malmstrom, Tanja Woyke, Roland Hatzenpichler, Victoria J. Orphan.

**Investigation:** Ranjani Murali, Hang Yu, Fabai Wu, Antoine Crémière, Rafael Laso-Pèrez, Roland Hatzenpichler, Victoria J. Orphan.

**Methodology:** Ranjani Murali, Hang Yu, Daan R. Speth, Fabai Wu, Antoine Crémière, Rafael Laso-Pèrez, Rex R. Malmstrom, Danielle Goudeau, Roland Hatzenpichler, Stephanie A. Connon, Victoria J. Orphan.

**Project administration:** Victoria J. Orphan.

**Resources:** Antoine Crémière, Rex R. Malmstrom, Tanja Woyke, Victoria J. Orphan.

**Software:** Daan R. Speth.

**Supervision:** Victoria J. Orphan.

**Validation:** Ranjani Murali.

**Visualization:** Ranjani Murali, Daan R. Speth.

**Writing – original draft:** Ranjani Murali, Kyle S. Metcalfe, Victoria J. Orphan.

**Writing – review & editing:** Ranjani Murali, Hang Yu, Daan R. Speth, Fabai Wu, Antoine Crémière, Rafael Laso-Pèrez, Rex R. Malmstrom, Tanja Woyke, Roland Hatzenpichler, Grayson L. Chadwick, Victoria J. Orphan.

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
