## [Editor Report · Decision Letter 0]

8 Dec 2022

Dear Dr. Murali, 

Please allow me to first apologize for the delay in the processing of your manuscript. I am sorry for this, and I thank you for your patience while I was discussing your manuscript with an Academic Editor. Thank you for submitting your manuscript entitled "Physiological adaptation of sulfate reducing bacteria in syntrophic partnership with anaerobic methanotrophic archaea" for consideration as a Research Article by PLOS Biology.

Your manuscript has now been evaluated by the PLOS Biology editorial staff, as well as by an academic editor with relevant expertise, and I am writing to let you know that we would like to send your submission out for external peer review.

Once your full submission is complete, your paper will undergo a series of checks in preparation for peer review. After your manuscript has passed the checks it will be sent out for review. To provide the metadata for your submission, please Login to Editorial Manager (https://www.editorialmanager.com/pbiology) within two working days, i.e. by Dec 10 2022 11:59PM.

Kind regards,

Paula

---

Senior Editor

PLOS Biology

---

## [Decision Letter · Decision Letter 1]

13 Feb 2023

Dear Dr. Murali,

Thank you for your patience while your manuscript "Physiological adaptation of sulfate reducing bacteria in syntrophic partnership with anaerobic methanotrophic archaea" was peer-reviewed at PLOS Biology. It has now been evaluated by the PLOS Biology editors, an Academic Editor with relevant expertise, and by several independent reviewers. 

In light of the reviews, which you will find at the end of this email, we would like to invite you to revise the work to thoroughly address the reviewers' reports.

As you will see below, the reviewers find your work interesting but they raise some concerns that should be solved before publication. Reviewer #2 thinks that the claims about physiology are not backed by experimental evidence and therefore, we consider that your conclusions should be tone down when talking about the functions deduced from genomes. Please, address the rest of the reviewers' concerns. 

Given the extent of revision needed, we cannot make a decision about publication until we have seen the revised manuscript and your response to the reviewers' comments. Your revised manuscript is likely to be sent for further evaluation by all or a subset of the reviewers.

**IMPORTANT - SUBMITTING YOUR REVISION**

*Re-submission Checklist*

*Published Peer Review*

*PLOS Data Policy*

Please note that as a condition of publication PLOS' data policy (http://journals.plos.org/plosbiology/s/data-availability) requires that you make available all data used to draw the conclusions arrived at in your manuscript. If you have not already done so, you must include any data used in your manuscript either in appropriate repositories, within the body of the manuscript, or as supporting information (N.B. this includes any numerical values that were used to generate graphs, histograms etc.). For an example see here: http://www.plosbiology.org/article/info:doi%2F10.1371%2Fjournal.pbio.1001908#s5

*Blot and Gel Data Policy*

Sincerely,

Paula

---

Senior Editor

PLOS Biology

REVIEWS:

Reviewer #1: Energy production anaerobic organisms.

Reviewer #2: Geobiology.

Reviewer #3: Microbial ecology.

Reviewer #1: This manuscript focuses on understanding the adaptations of syntrophic sulfate reducing bacteria (SRB) that make consortia with anaerobic methanotrophic archaea (ANME). They focus on four different clades of SRB that perform sulfate-coupled anaerobic oxidation of methane (AOM) - HotSeep-1, Seep-SRB2, Seep-SRB1a and Seep-SRB1g - and perform a genomic comparison with their nearest evolutionary neighbours. This is a very logical approach that indeed reveals several differences that are pertinent to the syntrophic associations and seem to be unique to each partnership and to have evolved independently. This is an interesting study that provides very relevant information, such as the recruitment of multiheme cytochrome proteins for DIET, the involvement of specific membrane complexes in the respiratory chain, some nutritional dependencies and the involvement of the extracellular matrix in the interaction.

I found the work to be interesting and carefully performed, and have only a few questions that I think merit further discussion:

- Why is it proposed that the Tmc complex reduces DsrC in the HotSeep-1, Seep-SRB1a and Seep- SRB1g, when all these organisms contain the DsrMKJOP complex, which is considered to be the physiological electron donor of DsrC. Why is a special focus put on Tmc? This is not clear throughout the manuscript.

- Also it is stated that Tmc is heavily adapted for DIET. Why??

- The TmcD of Seep-SRB1a and Seep-SRB1g appear to differ from that of other SRB (Figure S8). What is the difference?

- Also it is stated that Qrc is the major site of energy conservation in this respiratory pathway. What is the basis for this statement? I cannot tell why it should be the major site. Qmo and Dsr will also likely contribute, and possibly others.

- Please describe better what is the QmoC-fusion protein that also binds heme c. Is it also predicted to bind hemes b and FeS clusters like QmoC? And how many hemes c? From the description it does not seem to have proteins that would interact with DsrC, so the proposal that it transfers electrons from cytochromes c and DsrC does not seem justitified

- Lines 437 and 445: Why would electrons go from DsrC to NAD(P)H or ferredoxin? This does not make sense. Rnf or Mrp are not expected to interact with DsrC.

- A large abundance of HdrA-like proteins is also seen in non-syntrophic sulfate reducers and many other anaerobes. It is not clear if this is specific or relevant for the syntrophic ones.

Reviewer #2: When I got the invitation to review this manuscript titled "Physiological adaptation of sulfate reducing bacteria in syntrophic partnership with anaerobic methanotrophic archaea", I was excited to read about new discoveries on the physiology of this interesting syntrophic partnership from one of the world leading lab on the physiology of ANME/SRB consortia. However, after reading the paper I was surprised since the paper does not report any physiological experiments or data but is rather a predictive genomic analysis of existing (And some new) sequence data. But, predicting physiology from genomes is not the same thing as actual physiological studies. Thus, I found the way the manuscript is presented it is a bit misleading in this sense. The paper is very long, and reads for the most part like a review article rather than an original research article (for example, the manuscript has 115 citations). 

For most of the manuscript, the focus tends to be heavily on results of other studies that did physiology. Even expression analysis has already been done (refs 22, 24) and the manuscript seems to rely heavily on those earlier studies to make the points. This gives more the impression of a review article. These prior physiology studies are discussed at length and it seems like the manuscript tries to use those prior physiology studies to try and patch together a new story on physiology, using their genomic data. There has been a lot of papers on this general topic, and actually the same group published a paper last year in PLoS Biology that also used FACS sorting and genomes of ANME/SRB consortia. So, for me the impact of this new manuscript is a bit dliuted.

https://journals.plos.org/plosbiology/article?id=10.1371/journal.pbio.3001508

Here are just a few examples of how the papers title and conclusions are a bit misleading since there is no physiology reported in the paper (only predictions from genomes/metagenomes) :

The title starts with: "Physiological adaptation of sulfate reducing bacteria..."

line 139: "In this work, we described a physiological framework ..... "

line 146: "Our study explored the diverse physiological strategies that underlie..."

line 668 : "Physiological adaptation of syntrophic SRB to partnerships with ANME"

I think the manuscript has potential to become interesting, for example if the authors re focused it more onto the aspect of convergent evolution of the SRB into the syntrophy with the ANME. And, the varying steps of dependence during the development of the syntrophy like the loss of hydrogenase in the SRB as they become more dependent on electrons from the methane from the ANME (this would maybe be an interesting idea to for the authors to follow up on). The HGT story and the cobalamin evolution was also a potentially interesting pursuit. However, as interesting as those things are they are buried in a long manuscript that is full of speculations on physiology from genome data. All of these interesting evolutionary aspects are comparative genomics (not physiology) and in my opinion the manuscript would benefit from a major re-structuring and re-messaging towards the evolutionary aspects (and away from physiology predictions) and resubmitted to a more evolutionary focused microbiology specific journal.

Reviewer #3: This is a very long and quite well written paper describing genes, proteins and pathways present in syntrophic SRB that are involved in methane oxidation. The paper reads like a combination of a research paper and a review, drawing from lots of previous work to back up interpretations. I really enjoyed reading it and find the summaries, diagrams and interpretations helpful. It is often hard for me to tell how solid the data is supporting interpretations, but the authors tell a nice story and I am willing to go with it. 

Not a lot of comments considering how much information is here. 

Abstract is not very illuminating. What results were found that show mechanisms of syntrophy? This should be rewritten to include some more interesting results. Rather than spending most of the abstract on what was done. 

52-53. Not sure that I would use the definition allowing for less energy investment for each partner. Second part of the sentence is not correct. 

442-443. Reference is needed for this. 

517-526. This is really hard to follow. Should be shortened and smoothed out. 

566-568. Not sure what the point of this sentence is. Should be reworded or removed. 

622-636. This should focus on proteins present in the syntrophs and absent in the non syntrophs. Clearly, there are many components of biofilms, but this would be more interesting if the specific ones used for syntrophy are described and then others mentioned. 

650-666. this is good, but again should highlight mechanisms just present in syntrophs. 

SFIG16. Colored lines need to be described. 

681. Where are these trees? Authors should indicate where they are located here. 

696-697. I don't see how the latter point contributes to sequence divergence. 

700 Reword. 

724-725. I do not understand this sentence regarding the Observed diversity. 

731-733. this latter speculation is not necessary. Certainly possible, but no evidence.

---

## [Decision Letter · Decision Letter 2]

13 Jul 2023

Dear Dr. Murali,

Thank you for your patience while we considered your revised manuscript "Identification of physiological potential and evolutionary traits of diverse syntrophic sulfate reducing bacterial partners of ANME using comparative genomics" for publication as a Research Article at PLOS Biology. This revised version of your manuscript has been evaluated by the PLOS Biology editors, the Academic Editor and the original reviewers.

Based on the reviews, we are likely to accept this manuscript for publication, provided you satisfactorily address the remaining points raised by the reviewers. Please also make sure to address the following data and other policy-related requests.

1. DATA POLICY:

Regardless of the method selected, please ensure that you provide the individual numerical values that underlie the summary data displayed in the following figure panels as they are essential for readers to assess your analysis and to reproduce it: Figures 1, 2C, 4, 5, 7, and Supplementary Figures SF1, SF2, SF3, SF4, SF5, SF6, SF7, SF8, SF9, SF16, SF17, SF18.

2. Please note that sole deposition of data or code to GitHub would not be compliant with our policies, as this could be changed after publication (https://journals.plos.org/plosbiology/s/data-availability). However, once the data/code is final, you can archive your publicly available GitHub data to Zenodo. Once you do this, it will also generate a DOI number that you can provide us with. See the process for doing this here: https://docs.github.com/en/repositories/archiving-a-github-repository/referencing-and-citingcontent

3. Please provide a blurb which (if accepted) will be included in our weekly and monthly Electronic Table of Contents, sent out to readers of PLOS Biology, and may be used to promote your article in social media. The blurb should be about 30-40 words long and is subject to editorial changes. It should, without exaggeration, entice people to read your manuscript. It should not be redundant with the title and should not contain acronyms or abbreviations.

4. We suggest a change in the title: "Physiological potential and evolutionary trajectories of diverse syntrophic sulfate-reducing bacterial partners of ANME archaea" 

We expect to receive your revised manuscript within two weeks. 

*Published Peer Review History*

*Press*

Sincerely,

Paula

---

Senior Editor,

pjaureguionieva@plos.org,

PLOS Biology

Reviewer remarks:

Reviewer #1: The authors have clarified some points relative to my questions, but I some of the arguments are not really convincing and some bias is being introduced in the interpretations. An important point regarding energy conservation is that the authors are too focused on small differences and disregarding the fact that the two essential complexes for sulfate reduction, QmoABC and DsrMKJOP, are conserved in the syntrophic SRB, and are certainly the important sites for energy conservation (along with Qrc). This should be made more clear rather than focusing mainly on Qrc.

Role of the Tmc complex. The strong focus put on Tmc as the complex that reduces DsrC (lines 685-686) is in my view totally unfounded. It is based on the absence of TmcA, which happens only for SeepSRB-1g and SeepSRB-1a, whereas Hot-Seep1 has a full Tmc complex and no Tmc complex is present in SeepSRB2. It is possible that TmcA is encoded somewhere else in the genome in SeepSRB-1g and SeepSRB-1a, or is replaced by a different cytochrome c. On the other hand the argument that the absence of TmcA is a reflection of adaptation to syntrophic lifestyle or DIET does not make much sense because both Tmc and Qrc use the same electron donor, the TpIc3, which the authors propose is a amin palyer in periplasmic electron transfer also here, namely for Qrc. So, regardless of electrons coming from H2, formate or ANME they are likely transferred to TpIc3. So adaptation to syntropy does not need to imply loss of TmcA. So, I don't really see any evidence to support that Tmc will be more relevant to energy conservation in the syntrophic than in free living SRB, specially when there is DsrMKJOP, the bona-fide electron donor to DsrC, universally present in SRB. This is misleading and should be corrected.

QmoC-fusion protein. It is now clarified that this protein is a fusion protein that has hemes c and a CCG domain. This suggests to me that it is not really "QmoC"-like, and so maybe a different name should be given to it. QmoC has an integral membrane domain with hemes b (not c), and a cytoplasmic domain with regular FeS clusters as present in HdrC. The protein described seems to be a fusion of a cytochrome c with a CCG protein (HdrB family) that contain a non-cubane cluster.

Lines 748-749: There certainly has to be a way for electrons from ANME to reach ferredoxin and NAD(P)+, but I doubt this necessarily involves DsrC. This protein is specific for sulfur metabolism, and although it may connect with the ferredoxin and NADH pools through Flx/Hdr, it is highly likely that other proteins will be involved in the direct reduction of Fd and NAD(P)+, which are likely conserved in other organisms capable of DIET, but not sulfur metabolism (Rnf, Nfn, and others).

I also think the manuscript would gain from some reduction and streamlining of the text.

Reviewer #2: The authors have done a good job responding to my comments, it is appreciated that they clarify now that their data indicates physiological potential rather than actual experiments. As I pointed out in my first review, the genomic analyses of the syntrophic SRB are still interesting from an evolutionary perspective . The manuscript should have an impact, since as the authors point out many of the "SEEP" SRB clades have been historically mis-cited in the literature as syntrophic partners and this paper now shows clearly which of the subclades are syntrophic. The horizontal gene transfer analysis is also very interesting. I am satisfied with the changes to the manuscript and now recommend publication.

---

## [Editor Report · Decision Letter 3]

8 Aug 2023

Dear Dr. Murali,

Thank you for the submission of your revised Research Article "Physiological potential and evolutionary trajectories of syntrophic sulfate-reducing bacterial partners of anaerobic methanotrophic archaea" for publication in PLOS Biology. On behalf of my colleagues and the Academic Editor, Fengping Wang, I am pleased to say that we can in principle accept your manuscript for publication, provided you address any remaining formatting and reporting issues. These will be detailed in an email you should receive within 2-3 business days from our colleagues in the journal operations team; no action is required from you until then. Please note that we will not be able to formally accept your manuscript and schedule it for publication until you have completed any requested changes.

PRESS

Sincerely, 

Paula Jauregui

---

Senior Editor

PLOS Biology
